# DMAP: A Distribution Map for Text

**Tom Kempton**[*]
University of Manchester, UK
thomas.kempton@manchester.ac.uk

**Julia Rozanova, Parameswaran Kamalaruban, Maeve Madigan**
Risk and Security AI Lab, Visa Inc., UK
{yrozanov,kaparame,mmadigan}@visa.com

**Karolina Wresilo,**[*]**Yoann Launay**[*]
University of Cambridge, UK
kwresilo@hep.phy.cam.ac.uk, yl844@cam.ac.uk

**David Sutton & Stuart Burrell**
Risk and Security AI Lab, Visa Inc., UK
{dsutton,sburrell}@visa.com

## ABSTRACT

Large Language Models (LLMs) are a powerful tool for statistical text analysis, with derived sequences of next-token probability distributions offering a wealth of information. Extracting this signal typically relies on metrics such as perplexity, which do not adequately account for context; how one should interpret a given next-token probability is dependent on the number of reasonable choices encoded by the shape of the conditional distribution. In this work, we present DMAP, a mathematically grounded method that maps a text, via a language model, to a set of samples in the unit interval that jointly encode rank and probability information. This representation enables efficient, model-agnostic analysis and supports a range of applications. We illustrate its utility through three case studies: (i) validation of generation parameters to ensure data integrity, (ii) examining the role of probability curvature in machine-generated text detection, and (iii) a forensic analysis revealing statistical fingerprints left in downstream models that have been subject to post-training on synthetic data. Our results demonstrate that DMAP offers a unified statistical view of text that is simple to compute on consumer hardware, widely applicable, and provides a foundation for further research into text analysis with LLMs.

## 1 INTRODUCTION

A language model $p$ provides a wealth of information about a text $\underline{w} = (w_1 \cdots w_T)$ through the sequence of next-token probability distributions $p(\cdot | w_1 \cdots w_{i-1})$. We ask how to use this information to learn something about the text $\underline{w}$ or the language model $p$. To date, most efforts that use a language model to report statistical properties of a text use metrics that measure how unexpected each token is under $p$. A standard such metric is the average log-likelihood of each observed token,

$$\frac{1}{T} \sum_{i=1}^{T} \log p(w_i | w_1 \cdots w_{i-1}),$$

while a related but coarser variant, log-rank, replaces $p(w_i | w_1 \cdots w_{i-1})$ with the ordinal *rank* $r(w_i | w_1 \cdots w_{i-1})$ of the observed token in the descending list of next token probabilities. *Perplexity* is the exponential of the negative average per-token log probability.

---

[*]Work completed while at Featurespace, a Visa Solution.

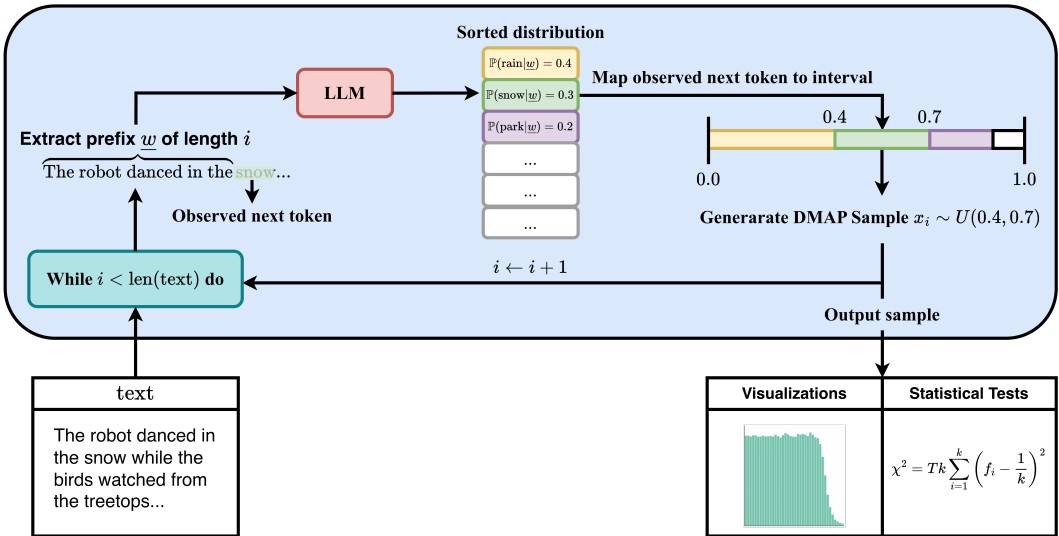

Figure 1: The DMAP algorithm. Given a text $\underline{w}$ of length $T$, this diagram illustrates how DMAP generates a collection of samples $x_1 \cdots x_T$ in $[0, 1]$. These may be analyzed quantitatively with statistical tests or qualitatively by splitting $[0, 1]$ into equal sized bins and plotting a histogram, as illustrated in Figure 2. We initialize $i = 1$. Our experiments demonstrate these visualizations identify decoding parameters (top-$p$, top-$k$, temperature), yield insights into black box machine-generated text detection algorithms based on probability curvature, and reveal statistical fingerprints left by performing supervised fine-tuning (SFT) on synthetic data.

Log-likelihood, log-rank and perplexity have been widely used in the training, evaluation, and detection of language models. For example, a body of work seeks to use perplexity to predict the readability of texts (Trott & Rivière, 2024), and to use the strength of the correlation between perplexity and human reading time as a measure of the quality of a language model (Oh & Schuler, 2023). However, in some settings these metrics are problematic (Meister & Cotterell, 2021; Fang et al., 2024), and often require contextualization to be useful.

In this work we use *contextualization* to describe the process of interpreting a raw statistic of a text (such as per-token log-likelihood) in terms of the content of the text. At the broadest level, one can give better answers to questions such as 'is a per-token log-likelihood score of 4 unusually high?' if one knows whether the text under consideration is a factual essay about chemistry or a piece of creative writing. A finer-grained approach is to compare the log-likelihood of a token present in a text to the expected log-likelihood of alternative tokens randomly sampled from a language model. The crux of the *contextualization problem* is that the way one should interpret a token $w_i$ being the third most likely, or having model probability $0.1$, depends on the number of reasonable token choices. These differences, encoded by the shape of the conditional distributions, are prone to persist over long passages of text.

Recently, a body of work initiated by DetectGPT (Mitchell et al., 2023) has tried to use a language model $p$ to address the contextualization problem and extract more nuanced information from next token probability distributions in the context of machine-generated text detection (Bao et al., 2023; Su et al., 2023; Hans et al., 2024). Inspired by these ideas, we introduce a distribution map, DMAP, which is statistically rigorous and lends itself better to visualization. In essence, DMAP is a recipe for mapping a text $\underline{w}$ onto a density function $f : [0, 1] \to \mathbb{R}^+$ that encodes information about both rank and perplexity in a principled way. As we shall see, this addresses the contextualization problem and provides a simple yet surprisingly powerful mathematical lens through which to measure and compare texts.

**Contributions**

1. **Introduce the DMAP Algorithm.** We introduce DMAP, a simple method to represent a text $\underline{w}$ through a language model $p$ as a set of samples in $[0, 1]$ that jointly encode rank

and probability information. DMAP is open-sourced[1], computationally efficient, and can be applied effectively on consumer hardware with small models such as OPT-125m (Zhang et al., 2022).

2. **Demonstrate Applications of DMAP.**

   (a) **Validate generation parameters.** Detecting incorrect or inconsistent generation settings in published data, such as top-$k$, top-$p$, temperature, or the language model itself.

   (b) **Design machine-generated text detectors.** Using DMAP, we identify design weaknesses of existing zero-shot detectors based on probability curvature and propose alternative principles for detector design.

   (c) **Reveal statistical signatures of post-training data in instruction-tuned models.** We investigate overconfidence in instruction-tuned models using DMAP and reveal alignment between downstream generated text distributions and fine-tuning datasets, even without access to internal model probabilities for the generating model.

We expect that the strengths of DMAP in statistical rigor and convenient visualization will see it find many applications beyond these.

## 2  RELATED WORK

**The Probability Integral Transform.**   The probability integral transform (PIT, Fisher 1928) was introduced as a method for mapping a continuous probability distribution on the real line onto the uniform distribution on $[0, 1]$, and as such is similar in philosophy to DMAP. One could combine PIT with the distributional transform to deal with non-continuous random variables, this corresponds exactly to our step of sampling from the uniform distribution on $I_i$. However, it only makes sense to use PIT with categorical variables if one has a natural ordering of the categorical variables, and if the bias which one is hoping to detect presents itself in terms of this ordering. While we have framed our work as turning log-rank into a statistically useful measure, one could instead think of DMAP as applying PIT and the distributional transform to the case of texts generated by language models, but with a crucial additional step of dynamic re-ordering tokens by model probability.

**Text Visualization.**   GLTR (Gehrmann et al., 2019) introduced a tool to color-code text according to token rank. This is an effective visualization, but inherits the issues of rank as a crude measure of where a token sits in the probability distribution. Our DMAP framework extends this visualization paradigm by providing a continuous, mathematically grounded representation that preserves both rank and probability information while enabling statistical inference.

**Machine-Generated Text Detection.**   Several key approaches have emerged to address the socially pertinent problem of detecting machine-generated text. DetectGPT (Mitchell et al., 2023) pioneered the use of probability curvature, measuring how perturbing a text affects its likelihood under a language model. This approach was refined by DetectLLM (Su et al., 2023), who adapted DetectGPT to use rank information, rather than exact probabilities. FastDetectGPT (Bao et al., 2023) addressed efficiency issues related to perturbation and the contextualization problem by normalizing probability distributions at each step. More recently, Binoculars (Hans et al., 2024) proposed a promising cross-model approach using probability ratios, though the theoretical justification for their normalization scheme remains unclear. Unlike these approaches, DMAP does not directly address the text detection problem. Instead, it provides a unified statistical framework that maps probability information to a standardized representation, enabling broader text analysis applications such as informing the design of future detectors.

**Model Calibration and Overconfidence.**   Recent work has identified systematic overconfidence in instruction-tuned language models. Luo et al. (2025) demonstrated that alignment procedures can degrade calibration, while Shen et al. (2024) proposed temperature scaling methods for post-hoc calibration. Chhikara (2025) investigated the relationship between alignment objectives and confidence estimation, and Zhu et al. (2023) analyzed calibration across different model scales. Yang & Holtzman (2025) explored how alignment training affects uncertainty quantification. DMAP

---

[1] https://github.com/Featurespace/dmap.

contributes to this literature by providing a tool to visualize and quantify distributional changes induced by post-training procedures, revealing statistical fingerprints that persist in downstream model behavior.

The key distinction of DMAP is its ability to map arbitrary probability distributions to a standardized unit interval representation, enabling both intuitive visualization and rigorous statistical analysis while addressing the fundamental contextualization challenges that limit existing approaches based on surprisal metrics.

# 3 DMAP: A DISTRIBUTION MAP FOR TEXT

## 3.1 DEFINING DMAP

Let $p$ be a language model, which we call the *evaluator model*, and let $\underline{w} = w_1 \cdots w_T$ be a text with tokens $w_i$ belonging to a vocabulary $V$. For each sequence position $i \in \{1, \cdots, T\}$, the candidate tokens $v \in V$ can be ordered by decreasing probability $p(v \mid w_1 \cdots w_{i-1})$. We construct a sequence of points $x_1 \cdots x_T \in [0, 1]^T$ referred to as a *DMAP sample* as follows, see Figure 1 for an overview.

Given a text $w_1 \cdots w_T$ and an index $i \in \{1, \cdots, T\}$, DMAP works by first defining an interval $I_i$ and then sampling a point $x_i$ from the uniform distribution on $I_i$. If $w_i$ is judged by the language model $p$ to be the most likely token to follow $w_1 \cdots w_{i-1}$, the interval $I_i$ will be $[0, p(w_i|w_1 \cdots w_{i-1})]$. More generally, $I_i$ is the interval of length $p(w_i|w_1 \cdots w_{i-1})$ whose left end point $a_i$ is the total mass of the set of tokens judged more likely by the language model.

Formally, let
$$V_i^+ := \{v \in V : p(v|w_1 \cdots w_{i-1}) > p(w_i|w_1 \cdots w_{i-1})\}$$
be the set of tokens judged by $p$ to be more likely than $w_i$ to appear next in the sequence.

Now, define two points $a_i$ and $b_i$ by
$$a_i := \sum_{v \in V_i^+} p(v|w_1 \cdots w_{i-1}),$$

and $b_i := a_i + p(w_i|w_1 \cdots w_{i-1})$.

Finally, define the interval $I_i \subset [0, 1]$ by $I_i := [a_i, b_i]$ and, for each $i$, let $x_i = D(w_i|w_1 \cdots w_{i-1})$ be chosen by sampling from the uniform distribution $U(a_i, b_i)$ on $I_i$, yielding the desired sequence $x_1 \cdots x_T$. In practice, to visualize the resulting set of samples $x_i$ we split the unit interval $[0, 1]$ into $k$ equally-sized bins and plot the resulting histogram. For the plots in this article we use $k = 40$. Other quantitative and qualitative measures of the set $\{x_i\}_i$ may be informative and we leave this for future work.

In the following proposition, pure sampling from language model $p$ refers to the practice of autoregressively generating a sequence $w_1 \cdots w_T$ by, at each step $i$, selecting token $w_i$ with probability $p(w_i|w_1 \cdots w_{i-1})$. We later also consider top-k, nucleus and temperature sampling, see Section B for definitions.

**Proposition 3.1.** *When generating a text $w_1 \cdots w_T$ by pure sampling from language model $p$, the corresponding sequence $x_1 \cdots x_T$ obtained by applying DMAP to $w_1 \cdots w_T$ with evaluator model $p$ will be independent and identically distributed (i.i.d.) according to the uniform measure on $[0, 1]$.*

*Proof.* See Appendix A. □

This proposition is particularly useful in Section 5.1, since the i.i.d. structure of the sequence $x_1 \cdots x_T$ allows one to use off the shelf results about convergence rates. In particular, it allows one to compute the chi-squared statistic of the distribution of the set of points $\{x_1, \cdots, x_T\}$, uncovering errors in the process of text generation with a high degree of confidence.

We can also use DMAP to visualize texts using an evaluator language model modified by a decoding strategy such as top-$k$, top-$p$ or temperature sampling. For example, to see how $w_1 \cdots w_T$ looks to language model $p$ at temperature $\tau$, one simply needs to replace the next token probabilities $p(v|w_1 \cdots w_{i-1})$ with the probabilities $q(v|w_1 \cdots w_{i-1})$ resulting from sampling from $p$ with temperature $\tau$. Proposition 3.1 continues to apply; see Appendix A.

### 3.2 DEFINING ENTROPY-WEIGHTED DMAP

There are two practical issues with the definition of DMAP in Section 3.1. First, there is randomness in the selection of $x_i$ from $I_i$ that introduces noise and makes DMAP non-deterministic. Second, if the language model $p$ has high certainty about the choice of next token, then a choice of $x_i$ contains little useful information and we should assign less weight to this choice of $x_i$. We present an alternative version of DMAP that mitigates these shortcomings by removing this randomness and weighting the outcome of each token by the entropy of the next token probability distribution. A version of Proposition 3.1 continues to hold, see Appendix A. In Appendix F we make DMAP plots restricted to times where the next-token probability distribution has low entropy, and see that these plots contain little useful information, justifying our deweighting of these points.

In broad terms, our map $\hat{D}$ defined below makes two changes to DMAP. Firstly, rather than weighting each sample equally, we use the entropy of the next-token probability distribution to weight each sample point, up to a chosen maximum clipped weighting of $\lambda$ for stability. Secondly, we mitigate the randomness in the selection of $x_i$ and, instead of considering the (weighted) proportion of samples in each bin, we plot the expectation of this, removing the randomness. A precise mathematical description goes as follows.

Given a language model $p$ and a text $w_1 \cdots w_T$, generate the sequence of intervals $I_1 \cdots I_T$ as with DMAP. Also, compute $h_1 \cdots h_T$ where

$$h_i := -\sum_{v \in \mathcal{V}} p(v|w_1 \cdots w_{i-1}) \log p(v|w_1 \cdots w_{i-1})$$

is the entropy of the probability distribution $p(\cdot|w_1 \cdots w_{i-1})$. Let $h_i' := \min\{h_i, \lambda\}$ and return the function $\hat{D}(\underline{w}) : [0,1] \to \mathbb{R}$ given by

$$\hat{D}(\underline{w}) = \hat{D}(w_1 \cdots w_T, p) := \frac{\sum_{i=1}^{T} h_i' \frac{\chi_{I_i}}{|I_i|}}{\sum_{i=1}^{T} h_i'},$$

where $\chi_{I_i}$ is the characteristic function equal to 1 on $I_i$ and 0 elsewhere, and $|I_i|$ is the length of interval $I_i$. This gives that $\chi_{I_i}/|I_i|$ is the function of integral 1 taking value 0 outside of $I_i$ and constant value on $I_i$. Thus, $\hat{D}$ is a step function of integral one supported on $[0,1]$. Finally, we may make an entropy-weighted DMAP plot by splitting the unit interval into, for example, $k = 40$ bins and averaging the value of $\hat{D}$ over each bin. Since the entropy is potentially very large, in practice we recommend clipping it at $\lambda = 2$ for stability.

The entropy $h_i$ can be viewed as the expected information obtained by revealing the next token $w_i$. Therefore, entropy-weighting places more weight on the times when the choice of next token is more uncertain, and so we have more to learn. This amplifies the differences between our output distribution and the uniform distribution at $[0,1]$, by reducing the weight associated with times $i$ where the next token is known with extremely high probability, making DMAP a more sensitive tool. Hereafter, we use entropy-weighted DMAP unless otherwise specified. See Appendix F for more information on entropy-weighting.

### 3.3 INTERPRETING DMAP VISUALIZATIONS

DMAP samples may be analyzed quantitatively, as in the $\chi^2$ tests of Section 5.1, or qualitatively via simple histograms. These visualizations reveal whether certain portions of the model next-token probability distribution are systematically over- or underrepresented in the text. Three particularly common behaviors that we observe are: **Head Bias** (e.g. Figure 2 (c)) - tokens viewed as likely by the evaluator model are over-represented in the text. **Tail Bias** (e.g. Figure 2 (f)) - tokens viewed as unlikely by the evaluator model are over-represented in the text. Text generated by one base model and evaluated by another base model with a similar entropy will typically display this behavior. **Tail Collapse** (e.g. Figure 2 (e)) - a small portion at the bottom of the evaluator model distribution is strongly under-represented in the text. Often seen in human-written text, this is consistent with the folklore intuition that model distributions place too much weight on tokens which are not realistic, an oft-cited motivation for top-$p$ (nucleus) sampling.

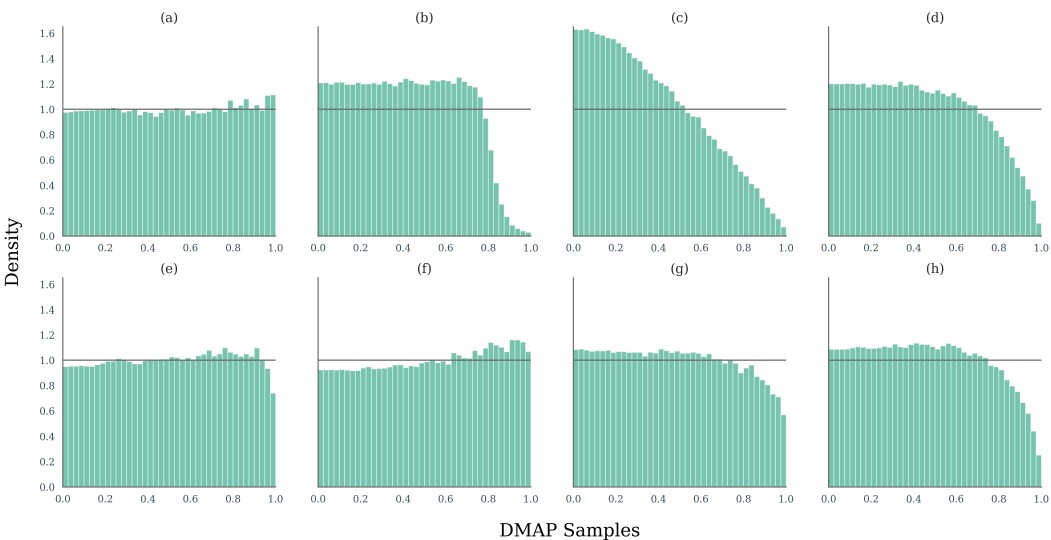

Figure 2: Illustrative DMAP histograms. The first row shows plots of XSum data (Narayan et al., 2018) generated by OPT-125m, evaluated by OPT-125m. The generation strategies (left to right) are (a) pure sampling, (b) top-$p$ = 0.8 sampling, (c) temperature $\tau = 0.8$ sampling, and (d) top-$k = 50$. The second row shows various different types of text evaluated by DMAP: (e) a news dataset of human text from RAID, (f) text generated by Mistral 7B (Albert Q. Jiang and others, 2023) using pure sampling (top-$p$=1), (g) text generated by Mistral 7B Instruct, and (h) text generated by ChatGPT from the Ghostbusters dataset (Verma et al., 2024). OPT-125m was used as the scoring model to generate DMAP samples. See Appendix N for examples with larger evaluation models.

## 4 ILLUSTRATIVE VISUALIZATIONS FROM DMAP

This section takes a first look at some histogram plots from $\hat{D}$. We take text from different sources, both human and machine, and evaluate them using OPT-125m as recommended by Mireshghallah et al. (2024). This evaluator demonstrates DMAP can be run effectively on consumer hardware in a few minutes. Throughout, we run DMAP over 300 texts each of around 300 tokens.

The top row of Figure 2 considers the case where the language model used to generate the texts is also used to generate the DMAP plot. We consider pure sampling (a), where token $w_i$ is selected at time $i$ according to its model likelihood $p(w_i|w_1 \cdots w_{i-1})$, along with top-$p$ (nucleus) (b), temperature (c), and top-$k$ (d) sampling. See Appendix B for an explanation of these decoding strategies. As predicted in Proposition 3.1, the DMAP plot from pure-sampled text with the same generator and evaluator language models is close to the uniform distribution. Temperature, top-$p$ and top-$k$ sampling are methods for increasing the probability of sampling from the head of the model distribution, corresponding to the left side of a DMAP histogram. They each produce head-biased DMAP plots with a highly characteristic shape. In particular, the plot for top-$p$ sampling is flat on $[0, \pi]$ before rapidly dropping off, and the plot for top-$k$ sampling is flat on roughly the first half of the interval before smoothly dropping off. These shapes can be explained in terms of the statistics of the space of conditional probability distributions; see Appendix B.

In the bottom row of Figure 2 we produce DMAP plots for text generated in various ways. Two things can be seen in the distribution of human-written text (e). The distribution generally shows that human-written tokens are somewhat surprising to OPT-125m. However, there is a sharp drop off on the very right hand side of the distribution, reflecting the fact that the bottom five percent of the OPT-125m distribution places too much weight on tokens which are not representative of human writing. This drop-off is much less pronounced when using more modern language models as detectors. Several other phenomena are easy to observe. Text generated by Mistral 7B (a base model) and evaluated by OPT-125m (f) is tail-biased. This means, on average, Mistral 7B generated text is surprising to OPT-125m. This phenomena should be expected when evaluating base models, see Section C, and is repeated when Mistral (Albert Q. Jiang and others, 2023), Falcon (Ebtesam

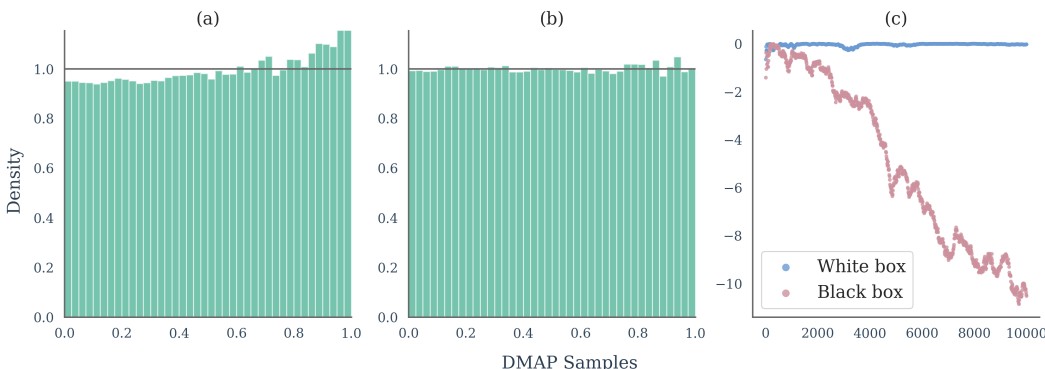

Figure 3: Quantitative validation of decoding parameters. (a) shows a DMAP plot for the black-box case of Llama 3.1 8B generated text evaluated by Mistral 7B. (b) shows a DMAP plot for the white-box case of Llama 3.1 8B generated text evaluated by Llama 3.1 8B. (c) plots the $log_{10}$ p-values resulting from our $\chi^2$ uniformity test. This demonstrates how quantitative evidence can be extracted from DMAP samples to investigate hypotheses about possible generation strategies. For example, (c) tells us that after evaluating 10000 tokens of Llama generated text with Mistral-7B as the evaluation model, the probability that Mistral would produce text with such an extreme $\chi^2$ distribution is less than $10^{-10}$. We can conclude that it is not plausible that the text under review was generated by pure sampling from Mistral 7B.

Almazrouei and others, 2023) and Llama 3.1 8B (Grattafiori et al., 2024) look at the text generated by one another, see Figures 8 and 9. In contrast to text generated by base models, text generated by instruction-tuned models Mistral-Instruct (g) and ChatGPT (h) is head-biased; on average these models are much more likely to pick tokens that are unsurprising to OPT. One might speculate that something in the post-training regime has profoundly altered the language model distribution. We study this question in more detail in Section 5.3.

## 5 APPLICATIONS

This section contains three example applications of DMAP, though we anticipate further use cases.

### 5.1 VALIDATING GENERATION PARAMETERS

Research claims in natural language processing are often sensitive to the precise decoding parameters used to generate text, see Section 5.2. Therefore, when generating text, or using publicly available samples, it is crucial to be able to validate the reported language model and generation hyperparameters such as top-$p$, top-$k$, or temperature.

For example, using DMAP, we discovered a major data-error affecting most of the top papers in zero-shot machine-generated text detection (Mitchell et al., 2023; Su et al., 2023; Bao et al., 2023; Hans et al., 2024; Dugan et al., 2024). At the time of their writing, HuggingFace enabled top-$k$ by default with $k = 50$, making it very easy for researchers to innocently and accidentally leave top-$k$ enabled while reporting that experiments were run on texts generated with pure sampling. Further works such as Dubois et al. (2025) use texts generated by other papers where the error was present. In this section, we present how DMAP may be used as a qualitative or quantitative tool to help researchers validate their data to prevent such occurrences.

**Qualitative Validation.** DMAP is an easy method for visually checking the integrity of machine-generated text for open-weight models. When text is claimed to be generated by pure sampling, Proposition 3.1 proves that the DMAP samples should be approximately uniformly distributed. Text generated by a given decoding strategy is clearly identified from the DMAP plots, see Figure 2. Alternatively, to test a specific combination of decoding parameters, we could test whether a uniform distribution is recovered when modifying DMAP to use the probability measure $q$ arising from sampling from $p$ with the specified decoding strategy.

**Quantitative Validation.** To go beyond a visual check, we have the following method for determining the consistency of the data with a reported generation method. Firstly, DMAP is applied to generate points $x_1 \cdots x_T$, adapted for the parameter settings, for example, by multiplying logits by $1/\tau$ for temperature sampling at temperature $\tau$. Next, the unit interval is divided into $k$ equal-sized bins, and frequencies $f_1 \cdots f_k$ computed, where

$$f_i = \frac{\#\{j \in \{1, \cdots T\} : \frac{i-1}{k} \le x_j < \frac{i}{k}\}}{T}.$$

We follow the Terrell-Scott rule (Terrell & Scott, 1985) for choosing the number of bins, letting $k = (2T)^{\frac{1}{3}}$ and compute the $\chi^2$ statistic

$$\chi^2 = Tk \sum_{i=1}^{k} \left( f_i - \frac{1}{k} \right)^2.$$

We then compute the probability that text generated by the reported generation method would have such an extreme $\chi^2$ statistic. This $p$-value determines the consistency of the text with the reported generation method, see Figure 3 and Appendix D for further details, and Goodman (2008) for guidance on the interpretation of p-values for this test.

## 5.2 Designing Methods to Detect Machine-Generated Text

This section describes how to use DMAP to gain insight into differences between human- and machine-generated text, and what conclusions can be drawn on the performance of current detectors and the design of future detectors. This is pertinent since, as discussed in Section 5.1, there are data errors in a substantial portion of the research on the detection of machine-generated text. Consequently, we discover that the principles underlying the design of many AI text detectors are not universally true, the primary being the *probability curvature* thesis. Other observations for the interested reader are provided in Appendix L.

### 5.2.1 Probability curvature for base and instruction-tuned models

The main idea behind most statistical approaches to detecting machine-generated text is that humans tend to choose words from further down the probability distribution than machines do. This idea was presented as *probability curvature* in DetectGPT (Mitchell et al., 2023), and the approach was broadly followed in DetectLLM (Su et al., 2023), Fast-DetectGPT (Bao et al., 2023), and Binoculars (Hans et al., 2024).

Probability curvature is effective in detecting text generated by an instruction-tuned model, or a sampling strategy that weights text generation towards the head of the distribution. However, the probability curvature idea is not supported by DMAP plots when pure sampling is used; in the white box case, Figure 2 (a) and (e), it is much weaker than previously reported and in the black box case, Figure 23, it appears false. To further validate this claim, we rerun experiments on the efficacy of DetectGPT, Fast-DetectGPT and Binoculars in the black box detection setting. When top-$k$ sampling is used the detectors remain effective as previously reported, since this decoding strategy impacts probability curvature as expected by the methods. However, Table 1 shows that when pure sampling is used their performance is worse than a coin toss due to inversion of the probability curvature. While an AUROC under $0.5$ may suggest inverting classifications on either side of the threshold, this is not possible here due to the zero-shot nature of these techniques: the fixed directionality is dictated by the method and based on the probability curvature thesis.

Table 1 shows that base models present an acute vulnerability to current detectors based on probability curvature. Fortunately, typical users will use instruction-tuned models which often exhibit head-bias as shown in Figure 2. However, new methods are required to prevent a determined adversary bypassing existing detectors through the use of base models with careful prompting. In addition, despite enterprise detectors most often being exposed to text from instruction-tuned models, the vast majority of the existing literature performs experiments on base models. Our results show these two classes of models should be tested separately when researchers are designing and validating future detection algorithms. Further experimental details may be found in Appendix J.

Table 1: Performance comparison across different machine-generated text black box detection methods, language models, and datasets (XSum (Narayan et al., 2018), SQuAD (Rajpurkar et al., 2016), and WritingPrompts (Writing, Fan et al. 2018)). Results are shown for two sampling configurations: top-$k = 50$ and pure sampling (top-$k = $ None). As predicted by DMAP, state-of-the-art detectors based on 'probability curvature' are effective when top-k sampling is used but not with pure sampled text.

| Method | Model | XSum | | SQuAD | | Writing | |
|---|---|---|---|---|---|---|---|
| | | $k = 50$ | $k = $ None | $k = 50$ | $k = $ None | $k = 50$ | $k = $ None |
| FAST-DETECTGPT | Llama-3.1-8B | 0.702 | 0.200 | 0.739 | 0.208 | 0.915 | 0.289 |
| | Mistral-7B-v0.3 | 0.770 | 0.276 | 0.819 | 0.299 | 0.906 | 0.339 |
| | Qwen3-8B | 0.765 | 0.289 | 0.612 | 0.320 | 0.923 | 0.377 |
| DETECTGPT | Llama-3.1-8B | 0.606 | 0.408 | 0.527 | 0.299 | 0.723 | 0.422 |
| | Mistral-7B-v0.3 | 0.679 | 0.486 | 0.586 | 0.365 | 0.688 | 0.457 |
| | Qwen3-8B | 0.635 | 0.445 | 0.463 | 0.380 | 0.724 | 0.479 |
| BINOCULARS | Llama-3.1-8B | 0.825 | 0.325 | 0.849 | 0.365 | 0.942 | 0.410 |
| | Mistral-7B-v0.3 | 0.823 | 0.350 | 0.851 | 0.416 | 0.931 | 0.404 |
| | Qwen3-8B | 0.857 | 0.416 | 0.752 | 0.467 | 0.949 | 0.492 |

## 5.3 STATISTICAL SIGNATURES OF POST-TRAINING DATA IN INSTRUCTION-TUNED MODELS

This section studies the effect of instruction fine-tuning on DMAP plots. In Figure 2 we saw that two instruction tuned models, ChatGPT and Mistral Instruct, systematically over-sample from the head of the OPT-125m probability distribution, whereas the non-instruction tuned base model Mistral 7B is tail-biased. This tail-biased behavior of the base model is understood and expected (see Appendix C). The real question is what is causing instruction tuned models to systematically over-select from tokens which OPT-125m finds likely?

It has previously been noted (Luo et al., 2025; Shen et al., 2024; Chhikara, 2025; Zhu et al., 2023; Yang & Holtzman, 2025) that instruction-tuned models are over-confident, in the sense that when answering questions they assign too much weight to answers they believe are likely correct. We hypothesize this may be an explanation for what we observe in our head-biased DMAP plots that place too much weight on likely tokens at the head of the distribution.

DMAP plots provide an indirect method for studying model overconfidence by checking for this head-bias in DMAP samples. We analyze whether this bias is present in the training data, the base model, and the corresponding fine-tuned model, thereby shedding light on how biases in training data may propagate to downstream model generations. This analysis is particularly relevant given the common practice of using temperature-sampled responses as fine-tuning data (Dubois et al., 2023).

We fine-tuned two sizes of Pythia models (Biderman et al., 2023) on the OASST2 dataset (Köpf et al., 2023) with responses provided by humans, Llama 3.1 8B at temperature 1, and Llama 3.1 8B at temperature 0.7. Figure 4 contains DMAP plots for texts generated by Pythia 1B with four fine-tuning configurations. For experimental details, DMAP plots for the fine-tuning data, and repeated results on the other model, see Appendix K.

Our main finding is that the only head-biased model was the one fine-tuned on temperature-sampled data, which was in turn the most head-biased fine-tuning data. We also see that human-written instruction fine-tuned data has a dramatic tail-collapse, much larger than seen in other human-written text. In addition, we see increased density in the final bin for fine-tuned models, which might demonstrate how DMAP could be used to detect mild overfitting during SFT and inform early-stopping strategies, a hypothesis we leave to explore further in future work. Finally, we note that all of our instruction-tuned models had a less tail-biased distribution than the original Pythia model. These initial experiments point to the utility of DMAP in studying how instruction-tuning affects the distributional properties of machine text produced by downstream models.

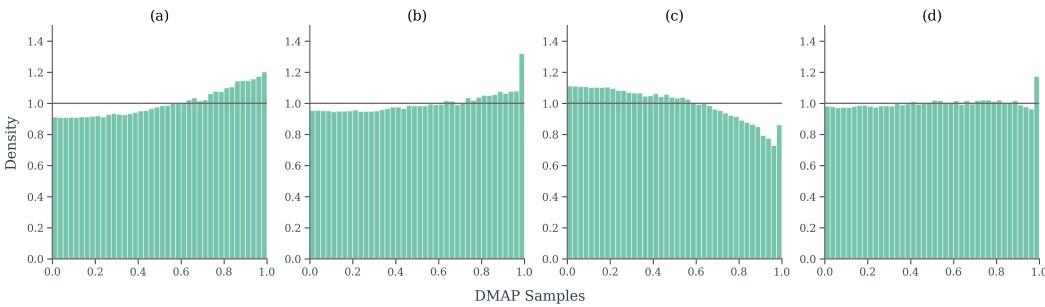

Figure 4: Using DMAP to investigate the effect of SFT post-training on synthetic data. DMAP plots generated with the evaluator model (OPT-125m) on text generated by Pythia 1B models with (a) no fine-tuning, (b) fine-tuned on OASST2 human data (Köpf et al., 2023), (c) fine-tuned on OASST2 with responses regenerated by Llama 3.1 8B at temperature 0.7, and (d) fine-tuned on OASST2 with responses regenerated by Llama 3.1 8B at temperature 1.0.

## 6    CONCLUSION

We introduced DMAP, a mathematically principled method for mapping text through language models to a standardized statistical representation. DMAP addresses the fundamental contextualization problem that has limited previous approaches to statistical text analysis with language models. Three initial case studies demonstrate the broad utility of this tool. Through parameter validation, we demonstrate how DMAP can validate data integrity, helping to prevent natural and inevitable human errors propagating through the research ecosystem. Our re-examination of detection methods showed that the widely-accepted *probability curvature* principle fails for pure sampling from base models, challenging foundational assumptions in the field and pointing towards principles for designing more robust detection strategies. Importantly, our experiments highlight that existing detectors are critically vulnerable to adversarial attacks using base models with pure sampling in the black box setting due to inversion of the expected curvature. Finally, our analysis of instruction-tuned models reveals how statistical fingerprints of training data persist in downstream model outputs, providing new insights into the sources of overconfidence in aligned models.

Beyond these specific applications, DMAP offers several key advantages: it is computationally efficient, requiring only forward passes through models such as OPT-125m that may be run on consumer hardware; it provides intuitive visualizations that make complex distributional patterns immediately apparent; and it enables rigorous statistical testing. The method's model-agnostic nature means it can be applied across different architectures and scales, making it a versatile tool for research and industry.

Our initial investigations point toward promising future research directions. For instance, DMAP might be used for data curation and efficient fine-tuning (Ankner et al., 2025), or to advance calibration approaches for instruction-tuned models, rather than using temperature scaling to align perplexity with pre-trained models. Alternatively, future work may calibrate DMAP distributions to match human text patterns, potentially offering more nuanced control over model confidence during frontier model training. The distinct DMAP signatures we observed across different text domains (poetry, news, technical writing) suggest the method could formalize how language model performance varies across contexts, separating inherent prediction difficulty from model-specific limitations. Perhaps most intriguingly, the dramatic differences in DMAP plots when generator and evaluator models differ point toward entirely new approaches to language model identification and forensic analysis. We are most excited for the applications we have yet to anticipate and believe that we have only scratched the surface of what we can learn through close examination of next-token probability distributions. DMAP provides a clear, principled window into that rich statistical landscape.

## 7    REPRODUCIBILITY STATEMENT

All datasets used are public or generated as detailed in our experimental setup in Section 5 and Appendices J and K. Code implementing DMAP is available on GitHub.[1]

## 8    ACKNOWLEDGMENTS

K.W. was financially supported by the Science and Technology Facilities Council (STFC) through the STFC DiS-CDT at the University of Cambridge. Y.L. was supported by the STFC DiS-CDT scheme and the Kavli Institute for Cosmology, Cambridge. We would also like to thank our reviewers for helpful feedback and comments.

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

## A   PROOF OF PROPOSITION 3.1

Suppose that a text $w_1 \cdots w_T$ has been generated by pure sampling from language model $p$. At time $i$, token $v$ is selected with probability $p(v|w_1 \cdots w_{i-1})$. Token $v$ defines an interval $[a, b]$ of length $p(v|w_1 \cdots w_{i-1})$ (see the definition of DMAP, Section 3.1). The points $a, b$ are functions of both $v$ and the context $w_1 \cdots w_{i-1}$, to keep notation clean we suppress this dependence here.

Then, following the DMAP algorithm, a point $x_i$ is selected according to the uniform distribution on $[a, b]$. Let $\mathcal{P}$ denote the partition of $[0, 1)$ into intervals $[a, b)$ corresponding to different tokens $v$ given fixed context $w_1 \cdots w_{i-1}$.

In order to prove that the process of picking $v$ according to $p(\cdot|w_1 \cdots w_{i-1})$ and then picking $x_i$ according to $U([a, b])$ produces points distributed according to $U([0, 1])$, it is enough to show that for any interval $(c, d)$, $\mathbb{P}(x_i \in (c, d)) = d - c$. It is enough to prove this for intervals $(c, d)$ which are fully contained in one of the intervals in the partition $\mathcal{P}$, the result for larger $(c, d)$ follows by standard rules of probability.

Now, given context $w_1 \cdots w_{i-1}$, let $(c, d)$ be a subset of some interval $(a, b)$ in $\mathcal{P}$, where $(a, b)$ is the interval corresponding to selection of some token $v$. Then

$$
\begin{aligned}
\mathbb{P}(x_i \in (c, d)) &= \mathbb{P}(x_i \in (a, b)) \cdot \frac{\mathbb{P}(x_i \in (c, d))}{\mathbb{P}(x_i \in (a, b))} \\
&= p(v|w_1 \cdots w_{i-1}) \cdot \frac{(d - c)}{(b - a)} \\
&= (b - a) \frac{(d - c)}{(b - a)} = d - c
\end{aligned}
$$

as required. Here we used that $x_i \in (a, b)$ exactly when the token $v$ defining interval $(a, b)$ was selected, which happens with probability $p(v|w_1 \cdots w_{i-1}) = b - a$.

We have shown here that, for any choice of context $w_1 \cdots w_{i-1}$, $x_i$ will be distributed according to $U(0, 1)$. Thus we have shown that $x_i$ is independent of $w_1 \cdots w_{i-1}$, and so the resulting sequence $x_1 \cdots x_T$ is i.i.d.

Finally we note that no properties of the language model were assumed in the above proof. In particular, one could define $p(\cdot|w_1 \cdots w_{i-1})$ to be the next token probability distribution resulting from sampling from Llama 3.1 at temperature 0.7. In that case, our result about $x_i$ being i.i.d. according to the uniform distribution on $[0, 1]$ continues to hold, provided the same language model decoding strategy pair is used in the generation of text and generation of DMAP plots.

Proposition 3.1 does not hold directly for entropy weighted DMAP, since entropy weighted DMAP does not produce points $x_i$. One can make corresponding statements about the expected integral of $\frac{\chi_{I_i}}{|I_i|}$ over any interval $A$ and see that entropy weighted DMAP plots for texts generated by the evaluator model do converge to the uniform distribution, but rates of convergence do not follow so easily.

## B   ON THE SHAPE OF DMAP PLOTS FOR TEMPERATURE, TOP-P, AND TOP-$k$ SAMPLING

Recall that, for a language model $p$ and context $w_1 \cdots w_{i-1}$, and for given values of $\tau$, $\pi$ and $k$, temperature sampling picks token $p_i$ with probability

$$
q_\tau(w_i|w_1 \cdots w_{i-1}) = \frac{p(w_i|w_1 \cdots w_{i-1})^{\frac{1}{\tau}}}{\sum_{v \in \mathcal{V}} p(v|w_1 \cdots w_{i-1})^{\frac{1}{\tau}}}.
$$

Top-$k$ sampling defines the top-$k$ set $\mathcal{V}_k$ to be the $k$ tokens with highest model (conditional) probability $p(\cdot|w_1 \cdots w_{i-1})$ and assigns probability

$$
q_k(w_i|w_1 \cdots w_{i-1}) = \frac{p(w_i|w_1 \cdots w_{i-1})}{\sum_{v \in \mathcal{V}_k} p(v|w_1 \cdots w_{i-1})}
$$

to tokens $w_i$ in the top-$k$ set, and zero mass outside of the top-$k$ set.

Finally top-p (nucleus) sampling orders the tokens in $\mathcal{V}$ by decreasing conditional probability $p(\cdot|w_1 \cdots w_{i-1})$ and defines the top-p set $\mathcal{V}_\pi$ to be the first $m$ tokens, where $m$ is the smallest integer for which $\sum_{i=1}^{m} p(v_i|w_1 \cdots w_{i-1}) \geq \pi$. Top-p sampling then chooses token $w_i$ with probability

$$q_\pi(w_i|w_1 \cdots w_{i-1}) = \frac{p(w_i|w_1 \cdots w_{i-1})}{\sum_{v \in \mathcal{V}_\pi} p(v|w_1 \cdots w_{i-1})}.$$

We then ask, given a long text sampled by top-$k$ sampling, what are the limiting statistics of the size of the top-$k$ set. The expected shape of the DMAP plot for top-$k$ sampling is the function

$$\mathbb{P}(\text{The total mass of the top-}k\text{ set is greater than }x)$$

renormalized to have integral one. This is a decreasing function, typically nearly flat on $[0, 0.5]$, before smoothly decreasing towards $0$.

Similarly, for top-$p$ sampling, the expected shape is the function

$$\mathbb{P}(\text{The total mass of the top-}\pi\text{ set is greater than }x),$$

again renormalized to have mass one. This is flat on $[0, \pi]$, before decaying very rapidly. There is a sharp cut off reflecting the majority of cases where the size of the top-$\pi$ set is only just larger than $\pi$, and a non-vanishing tail reflecting the fact that the proportion of times for which the top-$\pi$ set has mass close to 1 is non-zero.

For the shape of the DMAP plots in the case of temperature sampling, we mention only that it is a smooth deformation. There is an interval close to 0 upon which the DMAP is nearly flat, but this is much smaller than in the top-$k$ case as it is a function of the mass of the most likely token, rather than of the top-$k$ most likely tokens.

## C  WHY ARE BLACK-BOX BASE MODELS TAIL BIASED?

The fact that our plots of texts generated by one base language model and evaluated by another base language model (Figures 2 and 23) are tail-biased is partially supported by theory. In Kempton & Burrell (2025), it was shown that for a language model $P$ and a generation length $T$, pure sampling from $P$ is the unique way of maximizing the sum of entropy and per-token log-likelihood (as judged by $P$). That is, whenever we sample from a language model $Q$ for which the entropy is at least as large as the entropy of $P$, the resulting plot text must have lower expected per-token log-likelihood (as measured by $P$). This is a statement about log-likelihood, not a statement about position in the probability distribution, and so doesn't directly correspond to saying the DMAP plots should be tail-biased, but it gives a strong indication in that direction.

## D  RATES OF CONVERGENCE IN PROPOSITION 3.1

Proposition 3.1 implies that DMAP plots of texts generated by pure sampling from a language model, evaluated by the same language model, should look roughly flat. One might wonder whether we can make this statement more precise with a rate of convergence. The answer is positive, as detailed in the following proposition.

**Proposition D.1.** *Given a language model $P$, a text $w_1 \cdots w_T$ generated by $P$ and a number of bins $k$, let $x_1, \cdots x_T$ be the set of points generated by DMAP with $P$ as the evaluator model. Further define frequencies*

$$f_i = \frac{\#\{j \in \{1, \cdots T\} : \frac{i-1}{k} \leq x_j < \frac{i}{k}\}}{T}$$

*and the $\chi^2$ statistic*

$$\chi^2 = Tk \sum_{i=1}^{k} \left(f_i - \frac{1}{k}\right)^2$$

*as in Section 5.1. Then $\chi^2$ is asymptotically distributed according to the $\chi^2_{k-1}$ (the $\chi^2$ distribution with $k-1$ degrees of freedom).*

An immediate application of this is that we evaluate the plausibility of the statement 'Text $w_1 \cdots w_T$ was generated by pure sampling from language model $P$'. To do this, we compute the $\chi^2$ statistic $c$ arising from the text $w_1 \cdots w_T$ and use standard statistics packages to compute the probability that points drawn according to the $\chi^2_{k-1}$ distribution would be larger than $c$. For large $T$, this is arbitrarily close to the probability that text generated by language model $P$ would generate as extreme a $\chi^2$ statistic.

We mention one note of caution about the word 'asymptotically' in Proposition D.1. For finite $T$, the $\chi^2$ statistic is not perfectly distributed according to the $\chi^2$ distribution (indeed it is supported on a finite set), and so our computations of p-values are not exact. A conservative rule of thumb is that $T$ should be at least $10k$ for reliable $p$-values, so one should evaluate at least 400 tokens when plotting histograms with 40 bins.

Proposition D.1 is a standard result in statistics, proved by a slightly delicate application of the central limit theorem.

One should be cautious in interpreting p-values, as they are famously prone to misinterpretation. Many articles can be found explaining these misconceptions, see for example Goodman (2008).

## E EFFICIENCY AND EMPIRICAL CONVERGENCE RATES

This section provides a discussion on algorithmic efficiency and empirical results on the convergence of DMAP as the number of tokens, and therefore DMAP samples, grows. This latter evidence is to supplement the theoretical analysis of convergence in Proposition D.1. Figure 5 shows DMAP plots for 200, 2000, and 20,000 tokens. We see that for extremely small token sizes there is significant noise, while this quickly stabilizes. By 2000 tokens, we see identifiable patterns emerging, and by 20,000 the noise is fairly negligible with strong characteristic patterns present. If you are working with extremely small amounts of data with DMAP, you may reduce the impact of the noise by reducing the number of bins in your histogram. Even at 200 tokens this approach allows us to recover information about decoding strategies, see Figure 6.

The runtime efficiency and speed of DMAP is dominated by the underlying language model. For vanilla transformers, we therefore have quadratic time and memory complexity, though this could be improved if considering language models based on more efficient alternative attention mechanisms (Sun et al., 2025).

## F ENTROPY WEIGHTING

In Section 3.2 we introduced entropy weighting as a way of suppressing the effect of tokens selected at times where uncertainty around the next token is very low and a single option has extremely high likelihood (say probability 0.995). In such situations, the points plotted by DMAP tend to have distribution close to the uniform distribution irrespective of the method of text generation.

To illustrate this point, we consider DMAP plots of texts generated by Mistral-7B-v0.3 and evaluated by Llama-3.1-8B. The full entropy-weighted DMAP plot is present as the first plot on the second row of Figure 8. In Figure 7 we separate out points by the entropy of the next token probability distribution, plotting the low entropy case in (a), the medium entropy case in (b), and the high entropy case in (c). As predicted, in the low entropy case the DMAP plot is extremely flat, since very little information is uncovered from these points. This illustrates our reasons for giving lower weight to points of entropy below 2. In this example, tokens where the entropy is below $0.5$ make up 7474 of the 77321 tokens evaluated. In contrast, (b) and (c) contain noticeable peaks.

## G PROMPT SENSITIVITY ANALYSIS

This section includes experiments investigating the sensitivity of DMAP to the inclusion or exclusion of the prompt. In addition, we test the sensitivity of DMAP to the first few tokens for which there is no or small amounts of context to shape the conditional distributions.

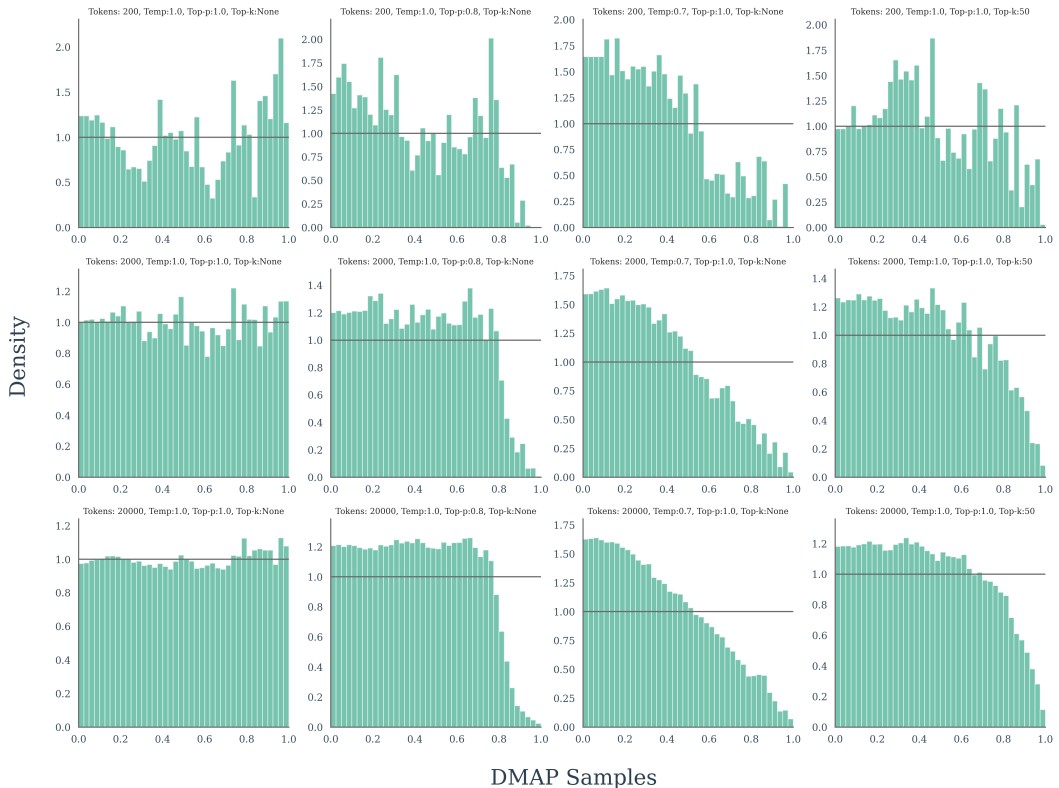

Figure 5: Empirical convergence analysis of DMAP with 40-bin histograms. This figure illustrates the convergence of 40-bin DMAP histograms as the number of tokens, and thus DMAP samples, increases from 200 to 20,000. At extremely low sample sizes we see high variability and noise, while the strong characteristic shapes we expect emerge as the number of tokens increases. We recommend choosing the number of bins as a function of the number of tokens being evaluated, for example using the Terrell-Scott rule, so as to mitigate this noise.

First, we repeated a large scale white box and black box experiment, to compare the efficacy of DMAP with and without access to the prompt. The results are shown in Figure 8 (without access to the prompt) and Figure 9 (with access to the prompt). These plots show that in realistic settings where one may wish to apply DMAP, the utility is equal with or without access to the prompt. However, note that without access to the prompt, despite the characteristic shapes being strongly present, there is a slight increase in the noisiness or irregularity in the plot. This is expected, since the reduced context will cause increased variance in the conditional distributions, particularly for tokens near the start of the text.

To explore sensitivity to the initial tokens further, for both settings including and excluding the prompt, we conducted an additional 32 experiments. Here, we also consider an *initial cutoff*. For an initial cutoff of $N$, we use the first $N$ tokens to compute conditional probabilities for tokens $N + 1, \ldots$ but do not plot the corresponding $N$ DMAP samples. This is intended to remove noise associated with the early tokens with insufficient context. We consider four core configurations i) include prompt and initial cutoff = 30, ii) include prompt and initial cutoff = 0, iii) exclude prompt and initial cutoff = 30, and iv) exclude prompt and initial cutoff = 0. Figure 10 analyses these four configurations for texts of length 300 tokens, while Figure 11 considers texts of length 50 tokens. These results show that at realistic token counts both the prompt and initial cutoff have only a minor effect. However, at very small token counts, DMAP is affected by the prompt. This is to be expected as before, since the prompt is a significant proportion of the total tokens in this case and therefore the statistical effect is larger. This gives a further reason to caution against using DMAP on extremely small samples of data.

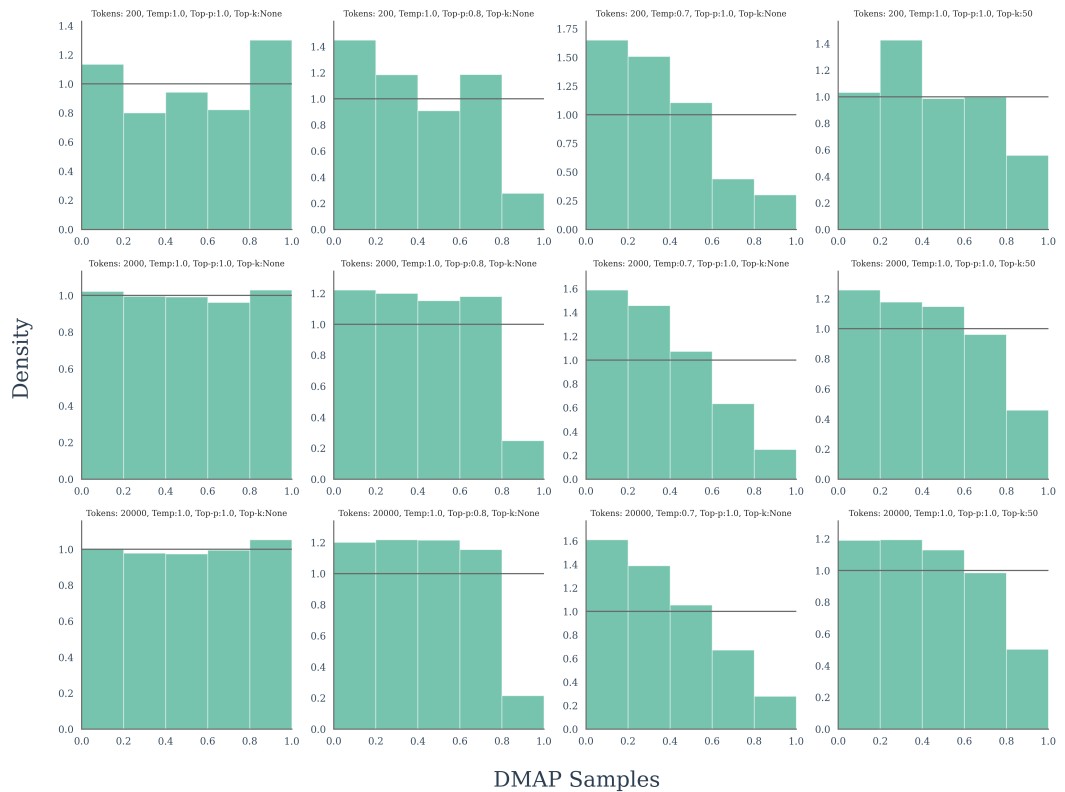

Figure 6: Empirical convergence analysis of DMAP with 5-bin histograms. This figure shows that noise due to extremely small sample sizes may be mitigated by using fewer bins. In this regime, we still see approximate versions of the characteristic patterns present in the top row of Figure 2, in contrast to the noisy estimates found in the first row of Figure 5.

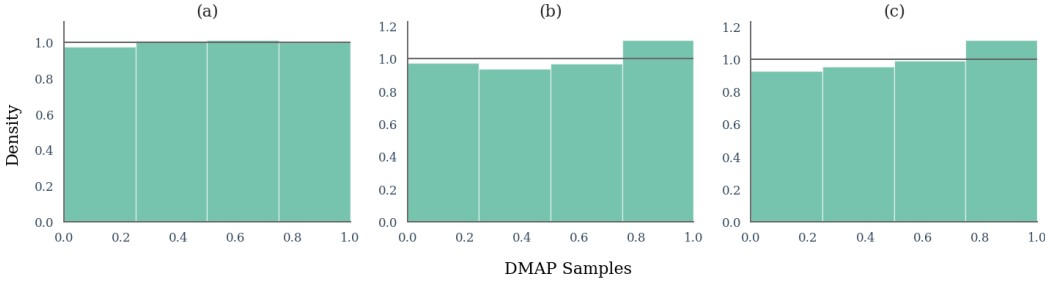

Figure 7: DMAP plots of text generated by Mistral and evaluated by Llama. Plot (a) shows only those tokens for which the entropy of the next token probability distribution is below 0.5, (b) shows entropy in the range (0.5,1), and (c) shows points where the entropy is greater than 2. We see that the plot (a) of low entropy tokens looks flat even though we are in the tail-biased case of pure sampling from a base model with different models used for generation and detection.

## H    RELATIONSHIP TO PROBABILITY INTEGRAL TRANSFORM

One way to view DMAP is that it builds on the classical probability integral transform (PIT), see Fisher (1928); David & Johnson (1948); Gneiting et al. (2007), as discussed in the related work section. However, a key difference between DMAP and PIT is that DMAP dynamically orders tokens by sorting the logits prior to generating a sample on the unit interval. As such, it also encodes *rank* information as studied in works such as Su et al. (2023). For completeness, in Figure 12 we

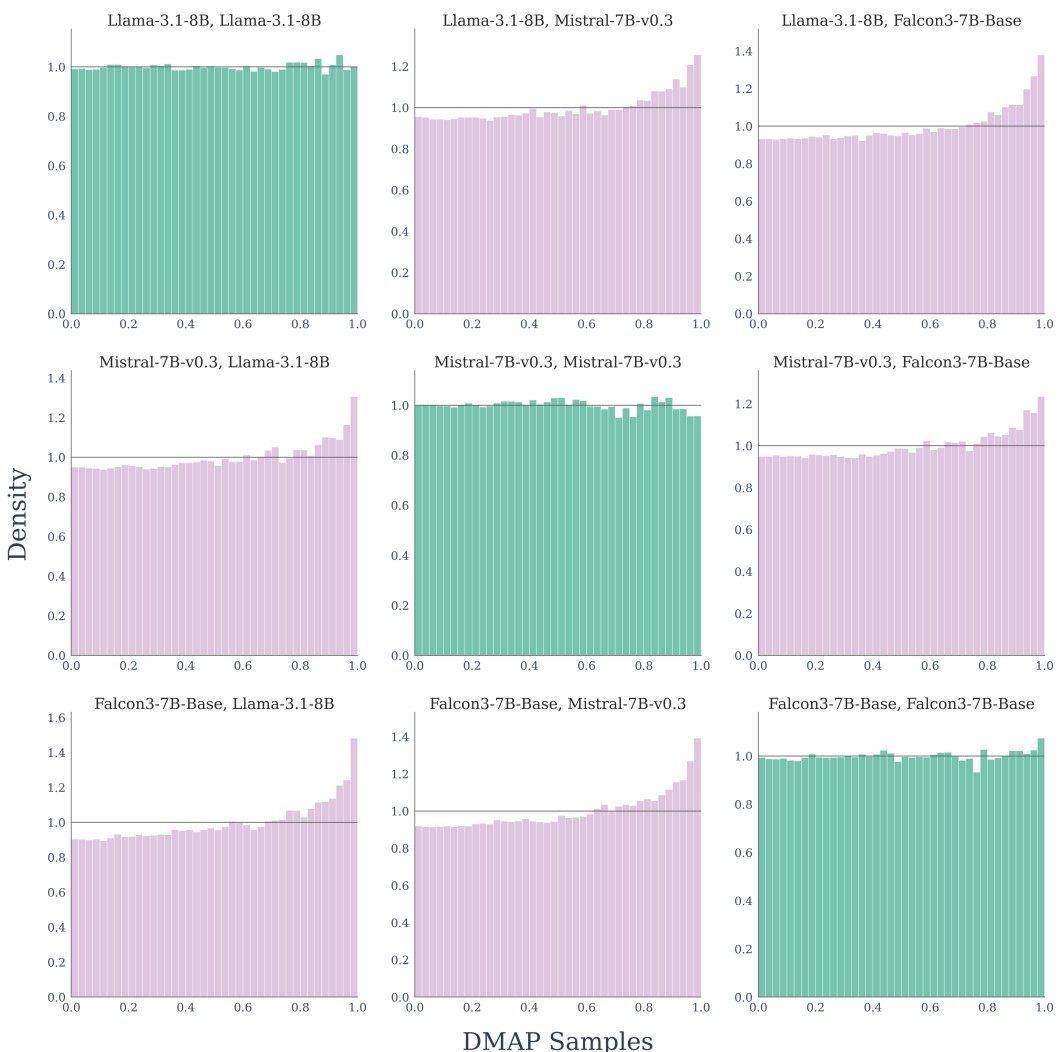

Figure 8: White and black box experiments excluding the prompt with texts of length 300. For these plots, the prompt was excluded. White and black box evaluation. Plots are labeled: (Generator Model, Evaluator Model). Plots on the main diagonal (green) correspond to the white box setting, while off-diagonal plots (pink) correspond to the black box setting. Observe that in the black box setting we see plots with heavy tails.

perform this extension of PIT to categorical variables where our dynamic token ordering is replaced with a random token ordering. We have been unable to extract useful information from these plots.

## I   ANALYSIS OF ADVERSARIAL PARAPHRASING

DMAP is a general tool for the statistical analysis of text, and not itself a specialized detector of machine-generated text. That said, we have seen throughout this paper that it can highlight important differences between machine and human generated text, for example, see Figure 2.

Paraphrasing is a well-known and important technique that has been shown to be an effective adversarial attack against common machine-generated text detectors (Sadasivan et al., 2025). In this appendix, we ask what the impact is of paraphrasing on DMAP visualizations. In particular, do paraphrasing attacks make DMAP visualizations of synthetic data look more like that of human data, analogous to their impact on common machine-generated text detectors? To answer this, we performed experiments using DIPPER (Krishna et al., 2024), a popular model used to evaluate ro-

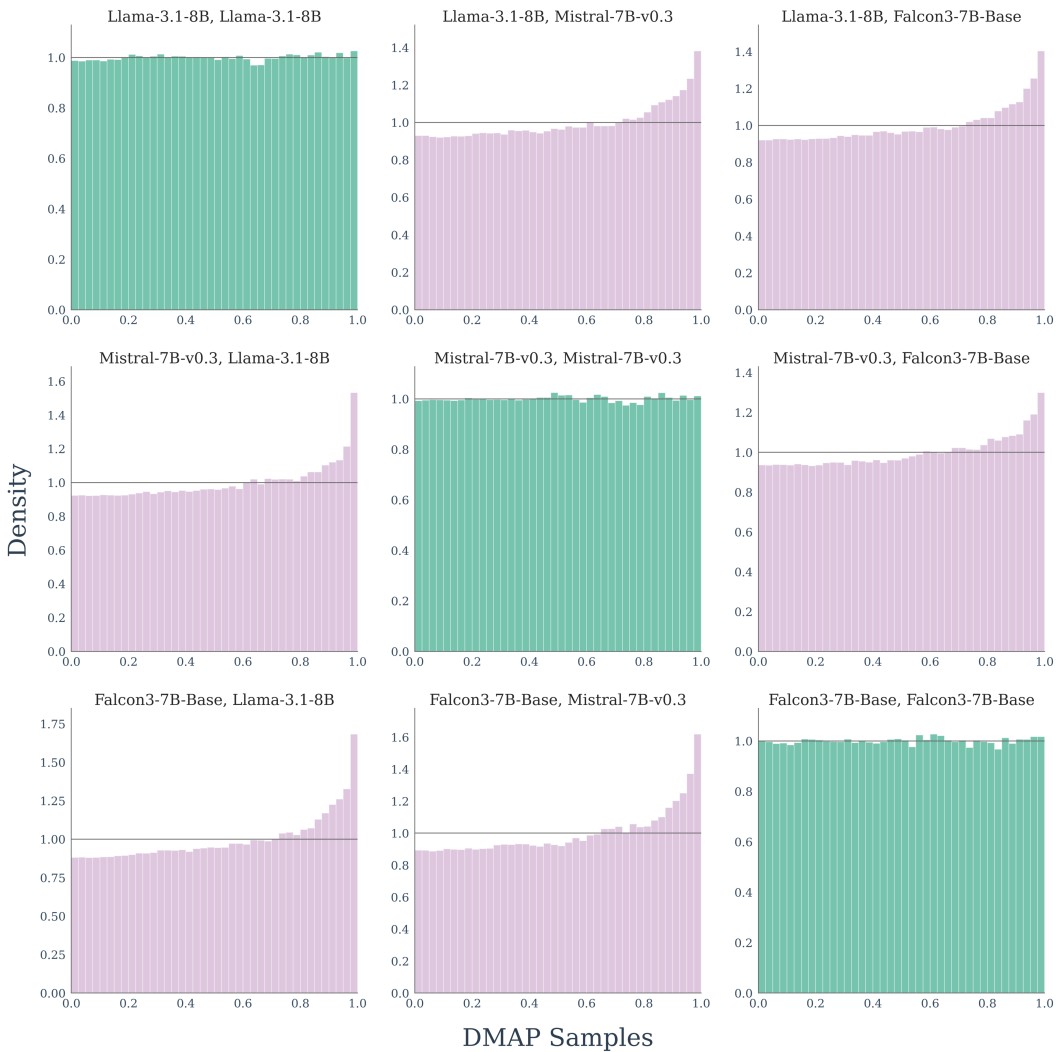

Figure 9: White and black box experiments including the prompt with texts of length 300. For these plots, the prompt was used in computation of next token probabilities of later tokens, but DMAP samples were not generated for tokens in the prompt. White and black box evaluation. Plots are labeled: (Generator Model, Evaluator Model). Plots on the main diagonal (green) correspond to the white box setting, while off-diagonal plots (pink) correspond to the black box setting. Observe that in the black box setting we see plots with heavy tails.

bustness to adversarial paraphrasing attacks, for example, see Sadasivan et al. (2025); Kempton et al. (2025). Our experiments show that DMAP visualizations are robust to paraphrasing attacks with DIPPER (Krishna et al., 2024). In particular, we see that paraphrased machine-generated text and human text are clearly distinct, see Figures 13, 14 and 15.

In addition, these figures also show that DMAP identifies subtle shifts between synthetic data and paraphrased synthetic data, shedding some new light on how paraphrasing attacks alter the distribution of text. Collectively, these results are strong evidence that an effective text detector may be built on top of DMAP that is far more robust to paraphrasing attacks than existing approaches. In addition, these results prompt the use of DMAP to explore the whole array of paraphrasing attacks in the literature to obtain a deeper understanding of their underlying statistical mechanisms.

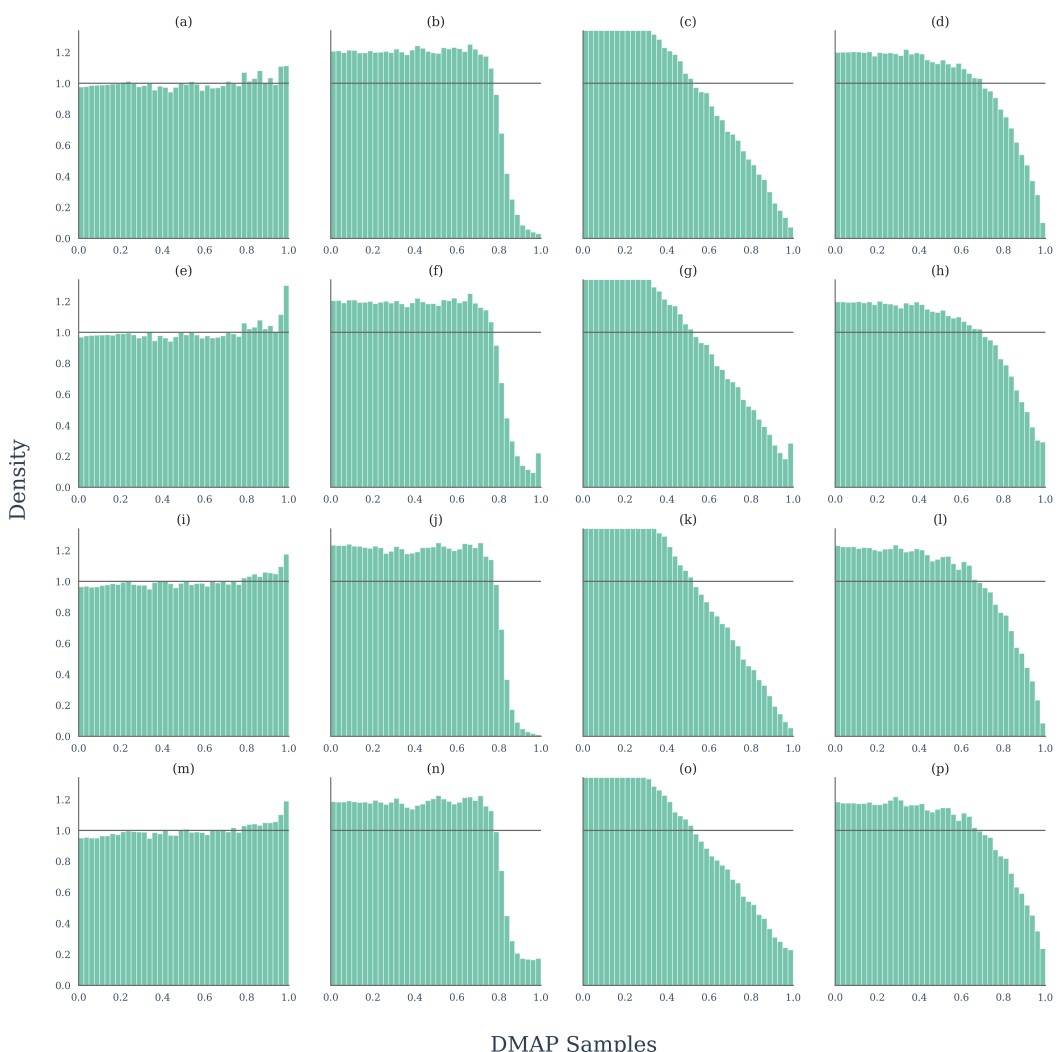

Figure 10: Prompt sensitivity analysis with texts of length 300 tokens. Each column corresponds to a generation strategy. From left to right columns, we have pure, top-$p$ = 0.8, temperature $\tau$ = 0.8, and top-$k$ = 50 sampling. (a)-(d) include the prompt and set initial cutoff = 30. (e)-(h) include the prompt and set initial cutoff = 0. (i)-(l) exclude the prompt and set initial cutoff = 30. (m)-(p) include the prompt and set initial cutoff = 0. The second and fourth rows show setting initial cutoff = 0 increased variability. Comparing the first and third rows shows excluding the prompt causes a slight increase in noise. In contrast to Figure 11, overall we see stable characteristic curves and low sensitivity to both the prompt inclusion and initial cutoff in this setting where the sample size is realistic. This is to be expected, since the prompt is a small proportion of the total tokens in this case.

## J   EXPERIMENTAL SETUP FOR SECTION 5.2

We follow a setup similar to, for example, Mitchell et al. (2023) or Kempton et al. (2025). In particular, given a large language model and sample of human text from XSum (Narayan et al., 2018), SQuAD (Rajpurkar et al., 2016), or WritingPrompts (Writing) (Fan et al., 2018), we construct 150 tokens of synthetic text using the first 30 tokens as context. We then perform zero-shot classification on the balanced dataset of real and synthetic data. For Fast-DetectGPT (Bao et al., 2023) and DetectGPT (Mitchell et al., 2023) we use GPT-Neo 2.7b (Black et al., 2021), which Fast-DetectGPT report as being empirically superior for black box detection with their method. For Binoculars we use the recommended combination of Falcon 7b for the observer model and Falcon 7b Instruct for

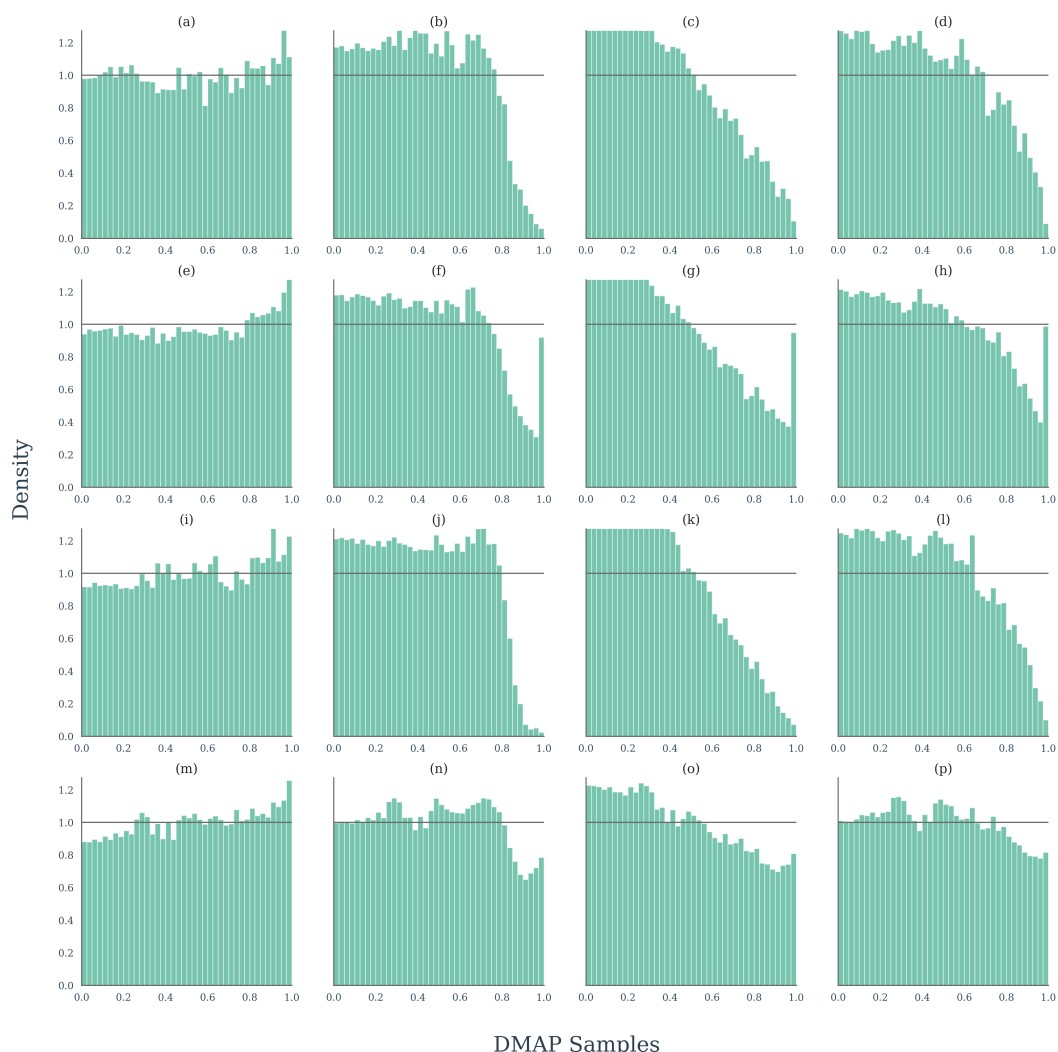

Figure 11: Prompt sensitivity analysis with texts of length 50 tokens. Each column corresponds to a generation strategy. From left to right columns, we have pure, top-$p$ = 0.8, temperature $\tau$ = 0.8, and top-$k$ = 50 sampling. (a)-(d) include the prompt and set initial cutoff = 30. (e)-(h) include the prompt and set initial cutoff = 0. (i)-(l) exclude the prompt and set initial cutoff = 30. (m)-(p) include the prompt and set initial cutoff = 0. All four rows show setting significantly increased variability and noise compared to the setting with 300 tokens per text (Figure 10). Compared to Figure 10, we increased sensitivity to both the prompt inclusion and initial cutoff in this setting where the sample sizes are very small. This is to be expected, since the prompt is a significant proportion of the total tokens in this case.

the performer model (Ebtesam Almazrouei and others, 2023). The generation models we use are Llama-3.1 8B (Grattafiori et al., 2024), Mistral-7B-v0.3 (Albert Q. Jiang and others, 2023), and Qwen3-8B (Yang et al., 2025). For each method, we report AUROC as the evaluation score as in (Mitchell et al., 2023; Bao et al., 2023; Kempton et al., 2025).

## K  EXPERIMENTAL SETUP FOR SECTION 5.3

We conduct our experiments on two sizes of Pythia models (410m, 1B) fine-tuned with one of three instruction fine-tuning datasets: one including only human-written text and two with responses that have been partially regenerated with an external language model.

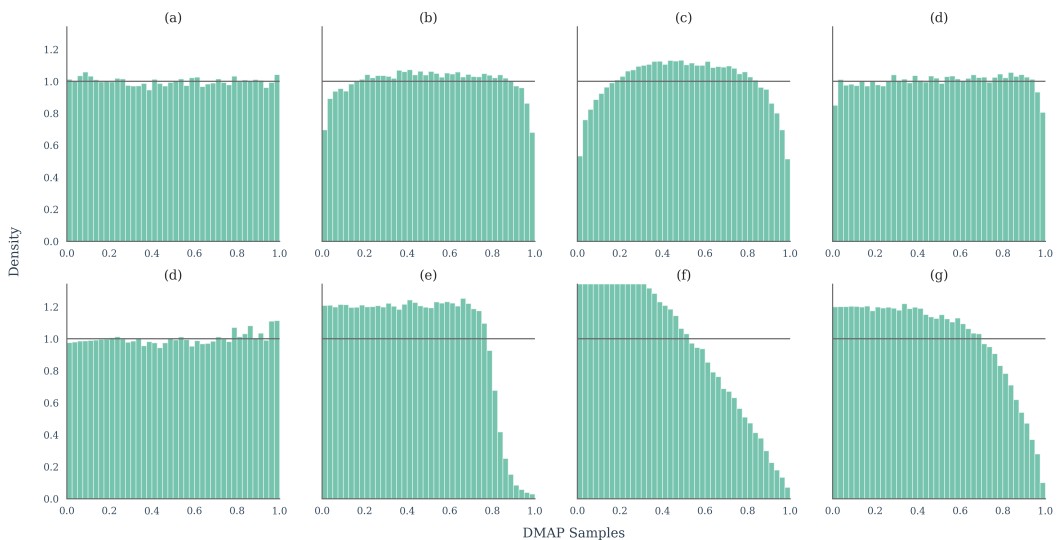

Figure 12: Comparison of DMAP with Probability Integral Transform. Each column corresponds to a generation strategy. From left to right columns, we have pure (top-$p$=1), top-$p = 0.8$, temperature $\tau = 0.8$, and top-$k = 50$ sampling. The top row (a)-(d) shows PIT with uniformly random token order. The bottom row (e)-(h) shows standard DMAP for comparison. In this scenario, we see plots for PIT with uniformly random token order do not effectively differentiate decoding strategies.

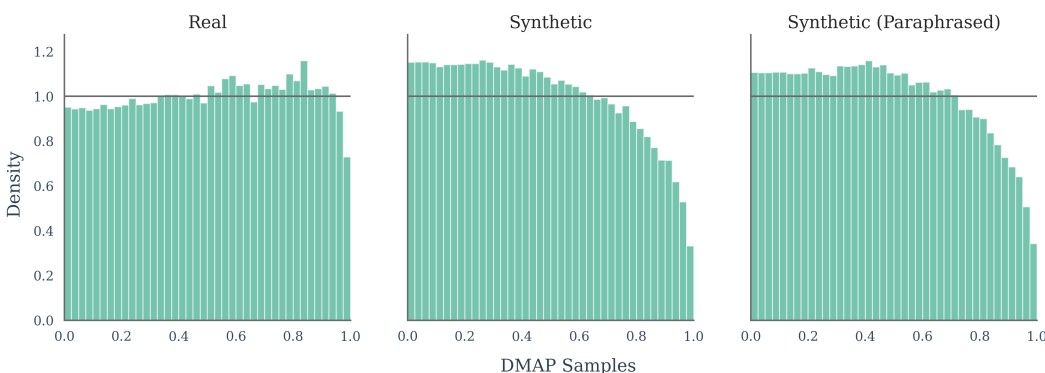

Figure 13: DMAP Analysis of Paraphrasing Attacks. These plots show DMAP visualizations for text generated by real, synthetic, and paraphrased synthetic data. The dataset used was SQuAD (Rajpurkar et al., 2016). Synthetic data was generated using Llama-3.1-8B completions. Paraphrased data was created using the model DIPPER developed for adversarial paraphrasing attacks on machine-generated text detectors (Krishna et al., 2024). These plots show that paraphrased machine-generated text and human text are clearly distinct in DMAP visualizations. In addition, DMAP sheds light on subtle changes in the distribution between standard synthetic and paraphrased text, where we see a slight flattening of the distribution.

**Fine-Tuning.** The instruction fine-tuning datasets are the following: (1) OASST2 (Köpf et al. (2023)), a human-written instruction tuning dataset, (2) OASST2 with responses regenerated with a Llama 3.1 8B model with generation temperature 0.7, and (3) The same as (2) but with generation temperature 1.0. OASST2 is a tree-structured multi-turn dataset with various conversation paths, but for our fine-tuning purposes we extract only the first prompt-response pair, as we only evaluate the response to a single query. Although this means we are using a portion of the text in the dataset, we continue to refer to it as OASST2 throughout this work. The data is split into a train and validation set following the original splitting of OASST2 on the Huggingface-hosted dataset [2]. Because the

---

[2] https://huggingface.co/datasets/OpenAssistant/oasst2

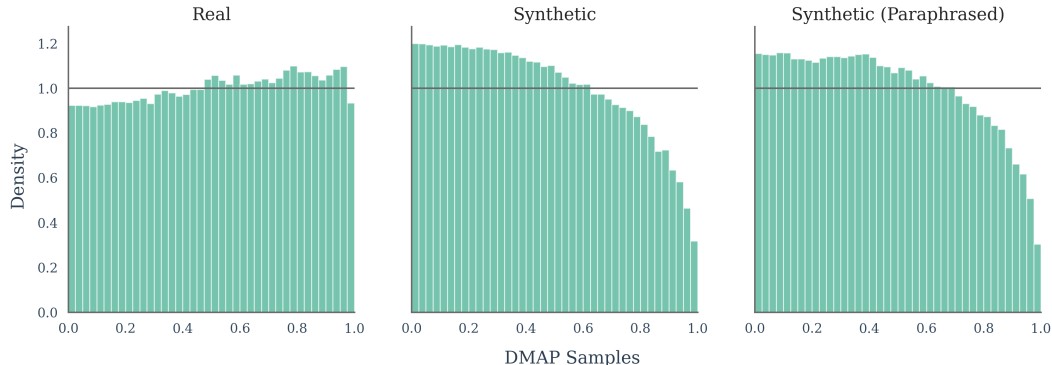

Figure 14: DMAP Analysis of Paraphrasing Attacks. These plots show DMAP visualizations for text generated by real, synthetic, and paraphrased synthetic data. The dataset used was Reddit Writing (Fan et al., 2018). Synthetic data was generated using Llama-3.1-8B completions. Paraphrased data was created using the model DIPPER developed for adversarial paraphrasing attacks on machine-generated text detectors (Krishna et al., 2024). These plots show that paraphrased machine-generated text and human text are clearly distinct in DMAP visualizations. In addition, DMAP sheds light on subtle changes in the distribution between standard synthetic and paraphrased text, where we see a slight flattening of the distribution.

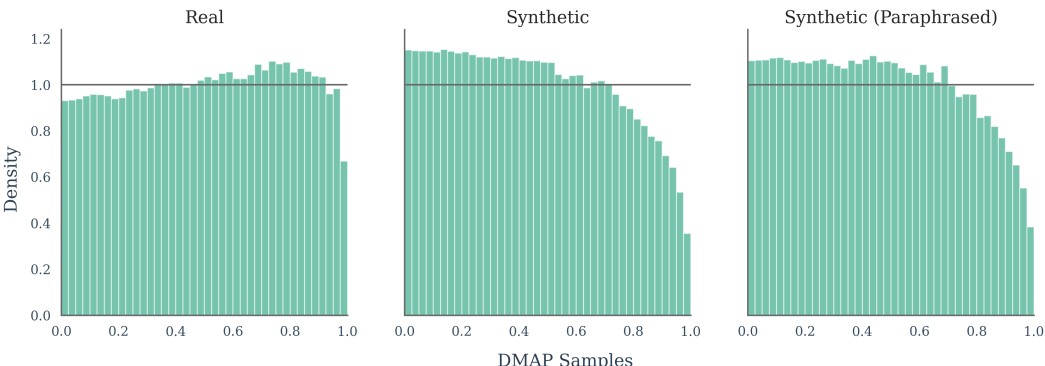

Figure 15: DMAP Analysis of Paraphrasing Attacks. These plots show DMAP visualizations for text generated by real, synthetic, and paraphrased synthetic data. The dataset used was XSum (Narayan et al., 2018). Synthetic data was generated using Llama-3.1-8B completions. Paraphrased data was created using the model DIPPER developed for adversarial paraphrasing attacks on machine-generated text detectors (Krishna et al., 2024). These plots show that paraphrased machine-generated text and human text are clearly distinct in DMAP visualizations. In addition, DMAP sheds light on subtle changes in the distribution between standard synthetic and paraphrased text, where we see a slight flattening of the distribution.

responses in fine-tuning datasets (2) and (3) are generated using a non-instruction-tuned base model, we keep the first 20 tokens of the human response text as part of the response generation prompt. For the creation of the DMAP plots, we only consider the tokens beyond this point.

For the fine-tuning step, the prompt-response pairs are formatted to be separated with a line break without any additional role tags or special tokens, so as to ensure a naturalistic output and to allow for exact comparison between fine-tuned and non-fine-tuned models.

**Evaluation.** We use OPT-125 as the evaluator model throughout, creating DMAP plots for the validation split of the respective datasets. In each case, the evaluated models are prompted in the same way as with the creation of the the training data, with only the newly-generated tokens contributing to the creation of the DMAP plot.

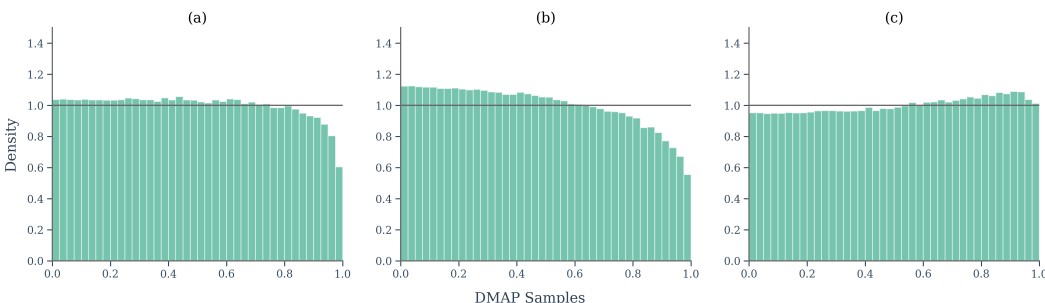

Figure 16: DMAP plots created using OPT-125 for the three fine-tuning datasets: a) OASST2 human-written prompt-response pairs, b) OASST2 with responses regenerated by Llama 3.1 8B at temperature 0.7. and c) OASST2 with responses regenerated by Llama 3.1 8B at temperature 1.0.

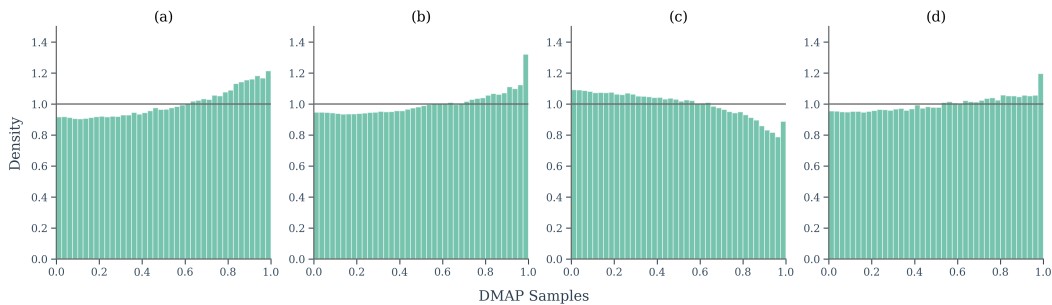

Figure 17: DMAP plots generated by pure sampling from Pythia 410m, models with (a) no fine-tuning, (b) fine-tuned on OASST2 human data, (c) fine-tuned on OASST2 with responses regenerated by Llama 3.1 8B at temperature 0.7, (d) fine-tuned on OASST2 with responses regenerated by Llama 3.1 8B at temperature 1.0.

### K.1 RESULTS

We first visualize the DMAP distribution for each of our three fine-tuning datasets in Figure 16, each of which exhibit a relatively typical shape for the respective data sources: human-generated, synthetic pure-sampled and synthetic temperature-sampled at temperature 0.7.

We observe next the extent to which fine-tuning on these datasets affects the DMAP plots. We present the results for a single model (Pythia 1B) in Figure 4, with the remaining experiment for Pythia 410m found here in Figure 17 showing the same visual pattern.

## L OBSERVATIONS ON DETECTOR DESIGN

### L.1 USING DIFFERENT DETECTOR AND GENERATOR MODELS CAN BE USEFUL

Methods based on probability curvature seem to work best when the generating and detecting models are the same (although Hans et al. (2024) and Dubois et al. (2025) use more than one model in detection). Comparing the distributions of human-written text (Figure 18), pure-sampled text where generator and detector models are the same (Figure 2) and pure-sampled text with different generator and detector models (Figure 23), one sees that human-written text looks most different from machine text when a different detector model is used. Perhaps an alternative conclusion to be drawn from Table 1 is that a variant Fast-DetectGPT can remain an effective detector of pure-sampled text, but DMAP should be first used to calibrate whether machine generated text is head-biased or tail-biased.

### L.2 Human-written text has a more subtle distribution than currently exploited

Detectors based on probability curvature exploit the fact that human-written text is tail-biased. In Figure 18 we see that, while this is true, it is also true that they contain tokens from the very tail of the language model probability distribution less often (tail-collapse). This is particularly important as the tail of the probability distribution is where log-likelihood values are at their most extreme. Detectors using log-likelihood or log-rank to distinguish between human and machine generated text would presumably be more effective if they discounted this bottom five percent of the probability distribution where human-written text is under-represented.

### L.3 Instruction Tuned Models are Easier to Detect than Base Models

It was noted that in Ippolito et al. (2020) that it is easiest for a machine to detect machine-generated text when decoding strategies such as top-p, top-$k$ and temperature have been used. Figures 2 and 18 would support this conclusion, the distributional differences between human and machine text are clearly much greater when top-$p$, top-$k$ or temperature sampling have been used.

## M   Further Questions

1. In Shen et al. (2024) the authors attempt to curb the overconfidence of instruction fine-tuned language models by using temperature scaling with temperature larger than 1 in order to bring the perplexity in line with pre-trained language models. Does DMAP give any insight to this process? What would happen if, instead of trying to bring perplexity in line with the pre-trained language model, one tried to bring the DMAP plots in line with human text? DMAP plots are a much better tool for checking calibration with a pre-trained language model than the currently used likelihood of the top token.

2. In Figure 18 we show DMAP plots for different examples of human-written text. These plots are not perfectly flat, and are slightly different in different settings (e.g. poetry vs news). Do these plots effectively describe the underperformance of the evaluator model in different settings in a way that separates the underperformance of the language model at next token prediction from the inherent difficulty of next-token prediction (which varies by setting)? Can one formalise this and extract useful metrics?

3. Figure 9 shows texts generated by Llama, Mistral and Falcon, evaluated by Llama, Mistral and Falcon. In each case, the shape of the DMAP plot is very different depending on whether the evaluator and generator models are the same. Can these ideas form the basis for a new approach to language model identification?

## N   Further Plots

### N.1   Synthetic Text

In this section we include illustrative plots with larger evaluation models. We see similar phenomena as in Figure 2.

### N.2   Human-written Text

We plot human-written text in Figure 22, evaluated by OPT-125m. We use the RAID dataset across four different categories: scientific abstracts, books, news and poetry.

We see differences in these plots, corresponding to the differing competence of OPT-125m in these areas.

### N.3   Black Box Base Language Models

In Figure 8 we plotted pure generated text from base language models Llama 3.1 8B, Mistral 7B and Qwen3 8B, evaluated by each other. In Figure 23 we see the same texts evaluated by OPT-125m.

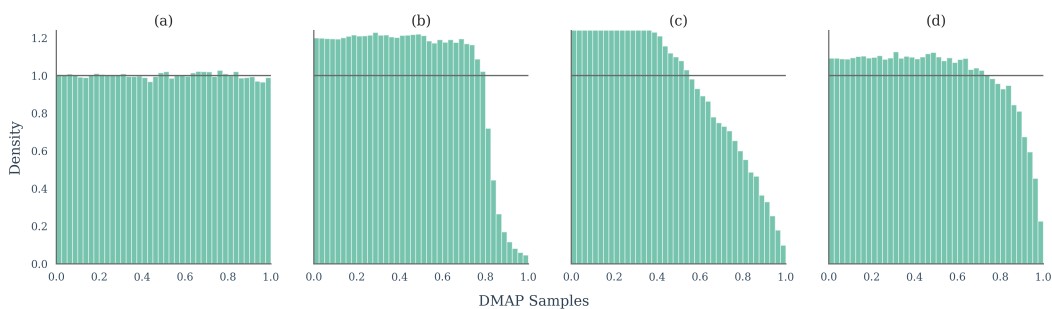

Figure 18: Illustrative DMAP histograms. Plots of XSum data (Narayan et al., 2018) generated by OPT-6.7b, evaluated by OPT-6.7b. The generation strategies (left to right) are (a) pure sampling, (b) top-$p$ = 0.8 sampling, (c) temperature $\tau = 0.8$ sampling, and (d) top-$k = 50$.

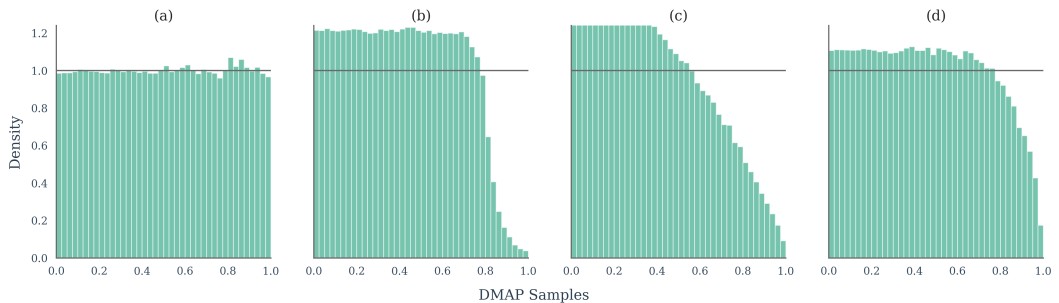

Figure 19: Illustrative DMAP histograms. Plots of XSum data (Narayan et al., 2018) generated by OPT-2.7b, evaluated by OPT-2.7b. The generation strategies (left to right) are (a) pure sampling, (b) top-$p$ = 0.8 sampling, (c) temperature $\tau = 0.8$ sampling, and (d) top-$k = 50$.

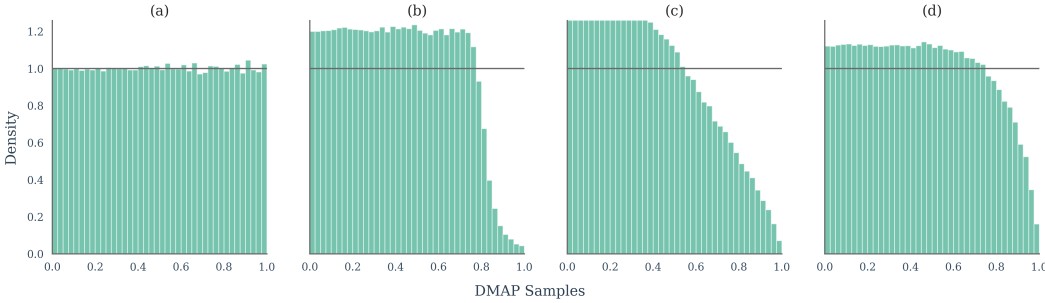

Figure 20: Illustrative DMAP histograms. Plots of XSum data (Narayan et al., 2018) generated by OPT-1.3b, evaluated by OPT-1.3b. The generation strategies (left to right) are (a) pure sampling, (b) top-$p$ = 0.8 sampling, (c) temperature $\tau = 0.8$ sampling, and (d) top-$k = 50$.

The images here further support the claim in the main text that pure-sampled text looks flat when the generator and evaluator models are the same, and looks heavy tailed (tail bias) when another base model is used for evaluation.

## O    LLM USAGE

LLMs were used to polish written text in the preparation of this manuscript.

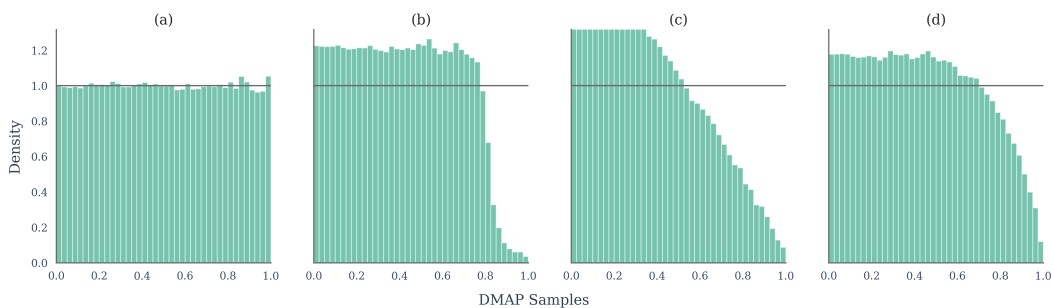

Figure 21: Illustrative DMAP histograms. Plots of XSum data (Narayan et al., 2018) generated by OPT-350m, evaluated by OPT-350m. The generation strategies (left to right) are (a) pure sampling, (b) top-$p = 0.8$ sampling, (c) temperature $\tau = 0.8$ sampling, and (d) top-$k = 50$.

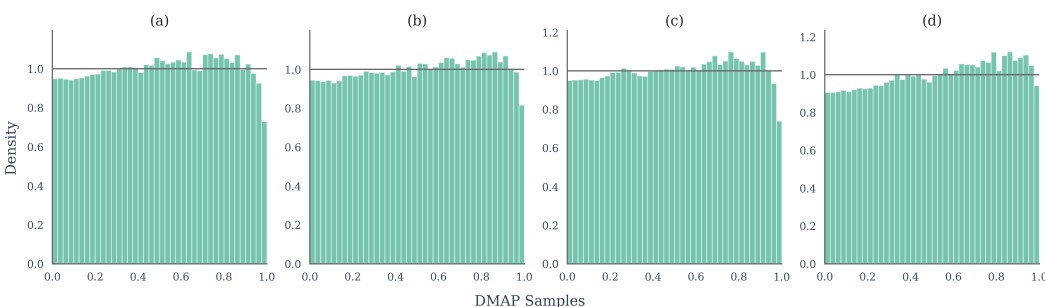

Figure 22: Human-written text in four categories: abstracts (a), books (b), news (c), and poetry (d).

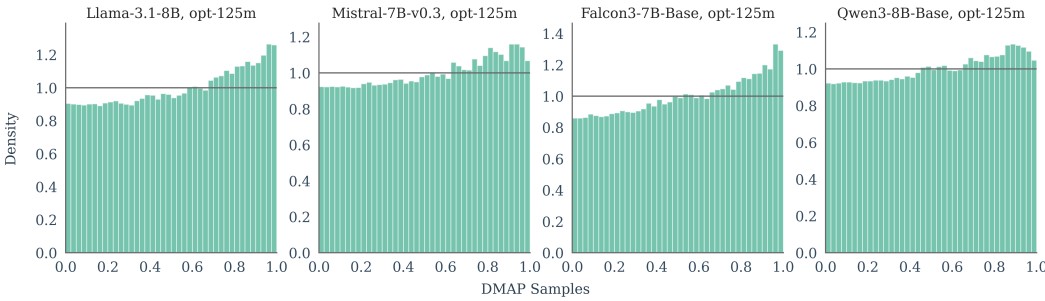

Figure 23: Llama 3.1 8B, Mistral 7B, Falcon 7b and Qwen3 8B generated XSum data, evaluated by OPT-125m.

