# OpenReview forum: "DMAP: A Distribution Map for Text"
_ICLR.cc/2026/Conference — ICLR 2026 Poster_

### Official Review · Reviewer_5pHM · 2025-10-30

**Soundness:** 3
**Presentation:** 3
**Contribution:** 2
**Rating:** 4
**Confidence:** 3

**Summary:**

This paper presents a straightforward method for visualizing a probability distribution on the unit interval which measures how the tokens in a piece of text relate to the token distribution of an evaluator language model. The visualization method has the property that if the same model M is used both to sample the piece of text, and to generate the visualization, then a uniform distribution should be produced. Furthermore, standard statistical tests of uniformity can be used to confidently reject the hypothesis that a given text dataset was produced by a particular model.

The authors then use this visualization method to point out an error in a significant line of work on detecting LLM generated text. In particular, this line of work inadvertently used top-k sampling when generating text, which clearly produces a bias towards more likely tokens. The property of bias towards more likely tokens was then proposed as a method for LLM generated text detection. The visualization method of the submission can be used to immediately see that this property is true for top-k sampling, but not for standard sampling. This implies that standard sampling can easily bypass these prior detection techniques. The method can also be used to as a straightforward statistical validation step, to verify the sampling model and settings (temperature, top-k, nucleus sampling etc) for any given text dataset.

**Strengths:**

The proposed method is straightforward and the visualizations are easy to understand. Most importantly, the method was used to identify a serious error in prior work on LLM generated text detection. The identification of the error alone seems to me a valuable contribution. I am not an expert in the LLM-generated text-detection research area, and so would defer to other reviewers if they have additional insight into the significance of this result.

**Weaknesses:**

Beyond the particular case of finding the error in prior work on LLM text detection, it is unclear how useful this method will be. In particular, the method is rigorous and clearly useful for saying that a particular text was *not* generated by an LLM, but it is unclear how robust this method is. For example, if the prompt used to generate the text is not available (as it generally is not when attempting to detect LLM generated text) it is unclear how this method will perform. The rigorous theoretical results do not hold without access to the prompt, and it is very unclear whether the empirical results will either.

**Questions:**

Does the method still at least give interesting empirical results without access to the prompt, but only the model response? Evidence in this direction would cause me to increase my score.

---

> ### Author Response · Authors · 2025-11-25
> **Review Response**
>
> Many thanks for sharing your insights and careful reading of the paper. We have added a significant new appendix based on your feedback:
>
> - Appendix G (Figures 8, 9, 10, and 11) contains a thorough analysis of whether DMAP is sensitive to inclusion of the prompt. In short, this appendix shows DMAP retains approximately equal utility with or without the prompt.
>
> Below we discuss each of your questions and comments in further detail.
>
> **Question:**
> Does the method still at least give interesting empirical results without access to the prompt, but only the model response? Evidence in this direction would cause me to increase my score.
>
> **Response:**
>
> This is a really useful question which lets us highlight one of the strengths of DMAP. For everything other than extremely short texts DMAP performs well without the prompts. This is because, for large N and for $m>N$, the difference between $p(\cdot|w_1\cdots w_{m-1})$ and $p(\cdot|w_{m-N}\cdots w_{m-1})$ is rather small, so the effect of tokens at least N steps in the past has a relatively small effect on the next token probability distribution. Theoretical results are rather thin on the ground here, so we have added a large additional appendix to thoroughly empirically validate this claim. Results can be found in Appendix G, Figures 8, 9, 10 and 11 in the updated manuscript.
>
> **Comment:** Beyond the particular case of finding the error in prior work on LLM text detection, it is unclear how useful this method will be. In particular, the method is rigorous and clearly useful for saying that a particular text was not generated by an LLM, but it is unclear how robust this method is. For example, if the prompt used to generate the text is not available (as it generally is not when attempting to detect LLM generated text) it is unclear how this method will perform. The rigorous theoretical results do not hold without access to the prompt, and it is very unclear whether the empirical results will either.
>
> **Response:** Firstly, on the general question of usefulness, we hope that the method for verifying that LLM generated texts presented in research papers were generated in the manner claimed will have applications well beyond the particular error that we discovered and will be a useful tool for reproducibility in language model research. More generally, we did not set out in this project to do the research outlined in our three applications (sections 5.1-5.3), rather, we made DMAP plots for various examples of language model generated text, found some of the plots surprising, and dug further into what was going on in those cases. At it’s core, DMAP is an easy and powerful method for spotting when outputs of a language model look a bit funny and are worthy of further scrutiny, we expect there to be many more things to learn.
>
> Thanks again to the referee for their comments, which have helped make some useful clarifications and important improvements to the paper.

---

### Official Review · Reviewer_soHE · 2025-11-01

**Soundness:** 3
**Presentation:** 2
**Contribution:** 2
**Rating:** 4
**Confidence:** 3

**Summary:**

The paper proposes DMAP, a simple yet effective algorithm that maps natural language text into a probability distribution on the [0,1] interval using a language model. DMAP provides an intuitive statistical visualization of how closely a text aligns with the model's generation configuration. The authors further demonstrate three possible applications of DMAP through concrete examples, supported by theoretical analysis or experimental evidence, to present its versatility and potential.

**Strengths:**

1. The paper introduces a clear and effective mapping algorithm that projects text into a probability distribution over [0,1] via a language model, enabling intuitive visualization and analysis.
2. The authors explore three potential applications of DMAP, each supported by either experimental results or theoretical analysis.

**Weaknesses:**

1. The core objective of DMAP is similar to the well-known Probability Integral Transform (PIT) used for model calibration [1–3]. Both map predicted probabilities to the [0, 1] interval and yield a uniform distribution when predictions are perfectly aligned with the model. DMAP can be interpreted as an LLM-specific extension of PIT that replaces i.i.d. sampling with conditional sampling and adds an entropy-weighting. The authors should explicitly discuss this connection and compare DMAP against a simple PIT-based baseline to clarify what additional insight DMAP provides in the context of LLM.

2. The paper claims that existing metrics such as log-likelihood and entropy fail to reflect "context" of the next-word prediction, yet these metrics already capture contextual information through the softmax normalization over the full next-token prediction's distribution. DMAP does not introduce new contextual information beyond this. Instead, it re-expresses and visualizes the same distributional information through a conditional sampling perspective. The authors should clarify how their notion of context differs from that already implicit in standard probability-based metrics.

3. DMAP effectively visualizes the inherent limitation of curvature-based AI-text detection methods: AI-generated text tends to be more "self-consistent" with the scoring model than human text, but this assumption breaks when the source model or decoding parameters differ substantially. While the paper illustrates this limitation, it does not demonstrate how DMAP may mitigate it as the authors claim in the paper. The authors should show how DMAP could improve detection robustness under cross-model and cross-config scenarios by applying DMAP to existing curvature-based AI-text detection methods empirically.

4. The paper does not specify how the reliability of DMAP visualizations changes with the input length. Since DMAP samples one value per token, short sequences may yield unstable or misleading results. The authors should provide theoretical or empirical guidance on the minimal input length required by DMAP, given the model and decoding parameters used.


[1] Fisher, Ronald Aylmer. Statistical methods for research workers. No. 5. Oliver and Boyd, 1928.

[2] David, F. N., and N. L. Johnson. "The Probability Integral Transformation When Parameters Are Estimated from the Sample." Biometrika, vol. 35, no. 1/2, 1948, pp. 182–90.

[3] Gneiting, Tilmann, Fadoua Balabdaoui, and Adrian E. Raftery. "Probabilistic forecasts, calibration and sharpness." Journal of the Royal Statistical Society Series B: Statistical Methodology 69.2 (2007): 243-268.

**Questions:**

Please check my questions in the Weaknesses section.

---

> ### Author Response · Authors · 2025-11-25
> **Review Response (1/2)**
>
> Many thanks to the reviewer for sharing your comments and insights. We have made several changes based on your comments, summarised below:
>
> - Added a dedicated paragraph in the related work section discussing the relationship of DMAP to Probability Integral Transforms (PITs).
> - Added Appendix H which contains a series of experiments empirically comparing DMAP to PITs in a variety of settings (Figure 12).
> - Added Appendix F which contains experiments illustrating the convergence rate of DMAP (Figure 5, Figure 6).
> - Clarified our definition of contextualisation in the introduction.
>
>
> **Comment:** The core objective of DMAP is similar to the well-known Probability Integral Transform (PIT) used for model calibration [1–3]. Both map predicted probabilities to the [0, 1] interval and yield a uniform distribution when predictions are perfectly aligned with the model. DMAP can be interpreted as an LLM-specific extension of PIT that replaces i.i.d. sampling with conditional sampling and adds an entropy-weighting. The authors should explicitly discuss this connection and compare DMAP against a simple PIT-based baseline to clarify what additional insight DMAP provides in the context of LLM.
>
> **Response:** Many thanks for introducing us to this beautiful line of work, of which we were unaware. We have updated our related work section to describe the relationship with PIT in detail, and have run a series of additional experiments comparing DMAP with PIT (Figure 12, Appendix H in the revised manuscript).
>
> Our motivation with DMAP was to try to modify log-rank to make it a more statistically useful measure, but having read the work on PIT one could also frame our work as using PIT with LLM next token distributions and entropy weighting, and crucially using token rank to dynamically order tokens. In the papers the referee linked to, PIT is used with continuous random variables on the real line. Extensions of PIT to discrete random variables do exist and use the same idea that we do of sampling a random point according to the uniform distribution on the interval in [0,1] corresponding to the observed output. However, for categorical variables, PIT is only useful for detecting bias relative to a given ordering. We computed plots using a random ordering of tokens rather than our dynamic reordering of tokens according to decreasing conditional probability, but were not able to extract useful information from these plots. They are included as Figure 12.
>
>
> **Comment:** The paper claims that existing metrics such as log-likelihood and entropy fail to reflect "context" of the next-word prediction, yet these metrics already capture contextual information through the softmax normalization over the full next-token prediction's distribution. DMAP does not introduce new contextual information beyond this. Instead, it re-expresses and visualizes the same distributional information through a conditional sampling perspective. The authors should clarify how their notion of context differs from that already implicit in standard probability-based metrics.
>
> **Response:** Thanks for this comment, we have added some extra clarification in the introduction. We mean by ‘contextualisation’ the process of normalising scores such as per token log likelihood in terms of the content of the text. This is a good thing to do since a per-token log-likelihood score of 3.5 might be considered surprisingly high for a chemistry essay but in line with expectations for a work of poetry. The Fast-DetectGPT algorithm gives a clever approach to contextualisation. We stress that this notion of contextualisation is not our idea, we just do it in a more statistically rigorous way.

---

> > ### Author Response · Authors · 2025-11-25
> > **Review Response (2/2)**
> >
> > **Comment:** DMAP effectively visualizes the inherent limitation of curvature-based AI-text detection methods: AI-generated text tends to be more "self-consistent" with the scoring model than human text, but this assumption breaks when the source model or decoding parameters differ substantially. While the paper illustrates this limitation, it does not demonstrate how DMAP may mitigate it as the authors claim in the paper. The authors should show how DMAP could improve detection robustness under cross-model and cross-config scenarios by applying DMAP to existing curvature-based AI-text detection methods empirically.
> >
> > **Response:** One comment we would make is that, even when the source model or decoding parameters differ substantially, in all the plots we've drawn we see DMAP plots of machine generated text look different from those of human text. So we don't expect the utility of DMAP in detecting machine text to be limited to the white box case. But the referee is absolutely right, we do not propose a solution to the problem of detecting machine generated text in this article (although some observations related to the problem are made in Appendix G). We will do so in a follow up article. Our reasons for this are twofold, firstly our intention in this article is to introduce DMAP as a visualisation tool with a variety of uses, and we suspect that if we focus too much on the problem of detecting machine generated text then the broad use cases we envision will become somewhat lost. Secondly, and more significantly, we just don’t have space in the present article to treat the problem of detecting machine generated text properly. The field is full of good ideas imperfectly implemented, with mutually incompatible results on the performance of different detectors presented in different papers. We need time and space to treat the problem of detecting machine generated text properly.
> >
> > **Comment:** The paper does not specify how the reliability of DMAP visualizations changes with the input length. Since DMAP samples one value per token, short sequences may yield unstable or misleading results. The authors should provide theoretical or empirical guidance on the minimal input length required by DMAP, given the model and decoding parameters used.
> >
> > **Response:** Thanks for this comment. We have added Figures 5 and 6 to the appendix looking at convergence rates for different numbers of tokens. Generally, rather than suggesting a minimum input length for stability, we suggest a maximum number of bins dependent on the number of tokens being evaluated (line 370, the Terrell-Scott rule). Further, for the white box case we discuss convergence rates in Section 5.1 / Appendix D.
> >
> > Many thanks again to the referee for their comments, which have helped us make useful clarifications and improvements to our article.

---

### Official Review · Reviewer_bpqg · 2025-11-01

**Soundness:** 4
**Presentation:** 3
**Contribution:** 4
**Rating:** 8
**Confidence:** 4

**Summary:**

This work presents DMAP that maps texts onto a distribution in the unit interval under a reference evaluating LLM, which provides both a direct visualization and quantifiable metrics for texts against the LLM, enabling identification of LLM-generated texts, validation of generation parameters, and the analysis of downstream LLMs.

**Strengths:**

1. The main proposal, DMAP, is well-grounded, intuitive, innovative, and immediately useful.

2. The effectiveness of DMAP is demonstrated through both comprehensive visualizations and analyses, and a good suite of downstream problems backed by solid experiments.

3. The authors points out a crucial problem with current state-of-the-art machine-generated text detection methods, and demonstrated DMAP's power in its identification.

**Weaknesses:**

To clarify, despite these weaknesses, I still think the paper has significant contributions to recommend acceptance. I'd like my review & subsequent discussions with the authors to work towards improving the paper quality.

1. As an antithesis to DMAP's ability to validate model sampling decoding strategies, DMAP is sensitive to decoding strategies and may not work as well if the temperature / top-K / top-p parameters change. This also may hinder DMAP's ability of becoming an effective tool for actual machine-generated text detection. I suspect there are ways to improve this, but it would require extensive additional experiments, especially more studies of DMAP on human-generated texts.

2. While the idea of DMAP becomes simple once understood, the paper could do a better job presenting the mathematical formulation. Here are some specific pointers:
    1. The introduction of $V_i^a$ and $V_i^b$ make the presentation bloated and unintuitive, and there should be some way around it. It may also be helpful if the intuitive explanation on Lines 155-157 are moved to the front, so the technical passage become easier to read (this is my recommended principle for writing in general).
    2. I fully understand the motivation of Section 3.2, and why it's a natural extension of the discrete, unweighted version of DMAP in Section 3.1. However, it will be helpful if, before the technical definitions are given, a quick explanation of the two things being done is given on Line 187, for example describing the extension of a single, uniformaly sampled point in an interval to a more "quantum" approach of keeping the interval intact as a step function, and the entropy weighting technique.
    3. Proposition 3.1 mentions pure sampling without first defining it.
    4. The example of temperature $\tau$ in Lines 172-176 can also end with "sampling from $p$ with temperature $\tau$" instead of "sampling from $p$ with the appropriate decoding strategy".

3. There's currently no analysis on the efficiency and sample complexity of DMAP, theoretical or empirical.

4. The visualization in Figure 1 is an important direct illustration that almost immediately explains the mechanism of DMAP. However it's too small to read and could use some more work.

**Questions:**

Q1. What, if anything, is stopping DMAP from developing into a tool for machine-generated text detection?

Q2. Why is the entropy directly used to weight the step functions at different tokens for Section 3.2's definition of DMAP? Did you consider other approaches rather than clipping, such as using the entropy of top-K tokens?

Q3. Figure 4 sees a heavy "tail" around the far end of the interval at 1.0. If I understand correctly, this corresponds to highly unusual tokens according to the validating LLM. Can you explain this further? Is this a common artifact of fine-tuning?

---

> ### Author Response · Authors · 2025-11-25
> **Review Response 1**
>
> Thank you very much for your careful reading of the manuscript and helpful comments. We have added some extra information in response to each of your points below and detailed the improvements to the manuscript that have been made based on your feedback.
>
> **Comment:** As an antithesis to DMAP's ability to validate model sampling decoding strategies, DMAP is sensitive to decoding strategies and may not work as well if the temperature / top-K / top-p parameters change. This also may hinder DMAP's ability of becoming an effective tool for actual machine-generated text detection. I suspect there are ways to improve this, but it would require extensive additional experiments, especially more studies of DMAP on human-generated texts.
>
> **Response:** Thanks for the comment, you are right that one should think carefully about whether different decoding strategies (or language models) might blunt the effectiveness of DMAP. For applications outside of detecting machine text, we view the sensitivity of DMAP to decoding strategies as a strength, not a weakness. It is this sensitivity that allowed our first application in verifying the generation parameters of LLM text and we hope this will be very useful in catching errors (or fraud) in research papers. In terms of detecting machine text, there is of course a concern that one might be able to bend a DMAP plot to look like a human distribution by a clever choice of generator model and temperature. One shouldn’t dismiss this possibility out of hand but we are skeptical that this method might prove effective against DMAP when the adversary does not know which language model is being used for detection.
>
> **Comment:**
> While the idea of DMAP becomes simple once understood, the paper could do a better job presenting the mathematical formulation. Here are some specific pointers ...
>
> **Response:**
> Thank you for these suggestions, we agree entirely and have implemented them all.
>
> **Comment:**
> There's currently no analysis on the efficiency and sample complexity of DMAP, theoretical or empirical.
>
> **Response:**
> Thank you for the comment. In terms of efficiency, the vast majority of the compute associated with running DMAP comes from running the text under evaluation through the language model to compute next token probability distributions. Ordering tokens by decreasing model probability is the next most compute intensive step, but is much quicker. We will add a more formal analysis of efficiency shortly.
>
> In terms of sample complexity, we have added Figures 5 and 6 to the appendix which shows DMAP plots for pure-sampling, top-k, top-p and temperature with varying numbers of tokens evaluated. We have theoretical results when evaluation and generation use the same model and decoding strategy, these are mentioned after the statement of Proposition 3.1 and used in Section 5.1 for our chi^2 goodness of fit tests.
>
> **Comment:**
> The visualization in Figure 1 is an important direct illustration that almost immediately explains the mechanism of DMAP. However it's too small to read and could use some more work.
>
> **Response:**
> We agree, and have expanded the plot and added extra details.

---

> > ### Author Response · Authors · 2025-11-25
> > **Review Response 2**
> >
> > **Comment:**
> > What, if anything, is stopping DMAP from developing into a tool for machine-generated text detection?
> >
> > **Response:**
> > We are very optimistic about using DMAP for machine generated text detection. But we believe it's a larger job than can be accomplished within the current manuscript, and so are working on a follow up question. Our reasons for postponing this work to a follow up article are twofold, firstly our intention in this article is to introduce DMAP as a visualisation tool with a variety of uses, and we suspect that if we focus too much on the problem of detecting machine generated text then the broad use cases we envision will become somewhat lost. Secondly, and more significantly, we just don’t have space in the present article to treat the problem of detecting machine generated text properly. The field is full of good ideas imperfectly implemented, with mutually incompatible results on the performance of different detectors presented in different papers. We need time and space to treat the problem of detecting machine generated text properly.
> >
> > **Comment**
> > Why is the entropy directly used to weight the step functions at different tokens for Section 3.2's definition of DMAP? Did you consider other approaches rather than clipping, such as using the entropy of top-K tokens?
> >
> > **Response:** This is an interesting question. Our motivation for capping the entropy at two was to preserve the stability (or speed of convergence) of DMAP by ensuring that most tokens are equally weighted, while suppressing those tokens with very low entropy. If all one is doing is making a visualisation, there's not an enormous amount of difference between these approaches, but once one has a concrete application in mind (such as detecting machine text) it certainly makes sense to play around with alternative weightings which might give stronger results.
> >
> > We have added Figure 7 to the appendix, where we confirm that plots consisting entirely of low entropy tokens do not contain much information.
> >
> > **Comment:** Figure 4 sees a heavy "tail" around the far end of the interval at 1.0. If I understand correctly, this corresponds to highly unusual tokens according to the validating LLM. Can you explain this further? Is this a common artifact of fine-tuning?
> >
> > **Response:** Thank you for your comment, your correct in your understanding that this corresponds to (a relatively small number) of tokens present in the text which appear highly unusual to the validating language model. We suspect that you are correct in that this is a common artifact of fine tuning. Since instruction tuning is very resource intensive we’re not able to verify this on larger models during the review period, but it’s an interesting hypothesis that we hope to return to.
> >
> > Thanks again for your careful reading of the manuscript and for your comments, which we believe have substantially improved the presentation of our work.

---

### Official Review · Reviewer_BUmF · 2025-11-02

**Soundness:** 2
**Presentation:** 3
**Contribution:** 2
**Rating:** 6
**Confidence:** 4

**Summary:**

The paper introduces DMAP, a mathematically grounded method to map next-token distributions from an LLM into samples on the unit interval, producing a unified representation that captures both rank and probability information. The authors show DMAP can (i) validate generation parameters, (ii) highlight weaknesses in machine-generated text detectors based on probability curvature, and (iii) reveal statistical fingerprints from instruction-tuning, especially when synthetic data is used.

**Strengths:**

- The paper is easy to read.

- The paper has discussed mathematical connections and also provided empirical results to support the claims made.

- This work provides some fresh ideas of going beyond perplexity while using LLMs to evaluate a given text.

- The paper presents nice visualization to explain the ideas.

**Weaknesses:**

- As a reviewer, I am confused about the problem setting in introuction, specifically the discussion of "contextualization
problem ", and why it is important.

- Following to that, the connection to AI generated text detection seems vague and not clearly established, and then authors introduce the DMAP algorithm without motivating for the need for it. This raises concerns about the novelty and not clear what is the key contribution of this work.

-  The authors talk about rannk being higher of v, but in the mathematical notation, it is lower? Am I missing something?

- The authors have mentioned about the limitaitons of DMAP and then proposed entropy weighted DMAP, this section discussion is confusing, is it possible to empirically highlight or show the limitations to make it clear what is being addressed with entropy weighted DMAP?

- As discussed in Table 1, it isnot clear why one would not use top-k sampling in practice? Ad what exact DMAP is predicting for these experiments?

- Can author comment more about the connection with uncertainty quantification literature?

**Questions:**

Please refer to the discussion in weaknesses.

---

> ### Author Response · Authors · 2025-11-25
> **Review Response**
>
> Many thanks to the reviewer for their careful reading of the manuscript and helpful comments.
>
> **Comment:** As a reviewer, I am confused about the problem setting in the introduction, specifically the discussion of "contextualization problem ", and why it is important.
>
> **Response:** Thanks for the suggestion, we have added some extra clarification in the introduction. We mean by ‘contextualization’ the process of normalising scores such as per token log likelihood in terms of the content of the text. This is a good thing to do since a per-token log-likelihood score of 3.5 might be considered surprisingly high for a chemistry essay but in line with expectations for a work of poetry. The Fast-DetectGPT algorithm gives a clever approach to contextualization. We stress that this notion of contextualization is not our idea, we just do it in a more statistically rigorous way.
>
> **Comment:** Following to that, the connection to AI generated text detection seems vague and not clearly established, and then authors introduce the DMAP algorithm without motivating the need for it. This raises concerns about the novelty and not clear what is the key contribution of this work.
>
> **Response:** Our key contribution is to introduce DMAP as a tool to visualize where a text sits (on average) in the next token probability distributions of a language model. We deliberately avoid motivating it purely in terms of our applications, because we expect the applications of DMAP to be much wider than those we have discussed. The relevance to AI text detection is in uncovering widespread experimental errors in the AI text detection literature (Section 5.1) and discovering that the widely believed ‘probability curvature’ hypothesis in AI text detection is false in many circumstances. We discovered these things by accident by drawing DMAP plots, and we think this is a strong argument in favour of DMAP being a very useful tool for the research community.
>
> **Comment:** The authors talk about rank being higher of v, but in the mathematical notation, it is lower? Am I missing something?
>
> **Response:** Many thanks for this comment and apologies for the confusion, which is bad terminology on our part (we made this mistake because ‘first’ is usually considered a higher rank than ‘second’, whereas ‘two’ is a higher number than ‘one’). We have amended the definition of DMAP to avoid this confusion.
>
> **Comment:** The authors have mentioned about the limitations of DMAP and then proposed entropy weighted DMAP, this section discussion is confusing, is it possible to empirically highlight or show the limitations to make it clear what is being addressed with entropy weighted DMAP?
>
> **Response:** Thank you for the comment and helpful suggestion, we have updated the main text, and further discussion and an empirical plot have been added (appendix F). In particular, we plot DMAP plots where we only consider low-entropy contexts, and find that the plots contain little information in that case, justifying the de-weighting of these points in the histogram.
>
> DMAP plots present information about where, on average, the tokens of a text sit in the next token probability distribution of a language model. We wish to weight this average so as to pay more attention to places where the choice of next token reveals more information. If one considers the sentence ‘The first ten whole numbers are 1,2,3,4,5,6,7,8,9,10’, very little information is contained in the final tokens, which reveal very little about the method of generating the text or the calibration of the language model. Entropy weighting reduces the influence of these tokens containing little information on the DMAP plot.
>
>
> **Comment:** As discussed in Table 1, it is not clear why one would not use top-k sampling in practice? And what exactly DMAP is predicting for these experiments?
>
> **Response:** A consequence of Table 1 is that one can avoid detection by state of the art machine generated text detectors simply pure sampling from a good language model. Given the substantial interest in evading detectors, we think this weakness of detection methods needs highlighting.
>
> **Comment:** Can author comment more about the connection with uncertainty quantification literature?
>
> **Response:** We mention uncertainty quantification in the related work section, in relation to model calibration (misaligned models are bad for uncertainty quantification). Our connection to this literature comes in Section 3, where we demonstrate a link between temperature sampled instruction fine tuning data and overconfidence in instruction tuned models. But we do not directly have anything to say with uncertainty quantification, it is that both our work and some work on uncertainty quantification take a look at the question of model calibration.
>
> Thanks again for your careful reading of the manuscript and helpful comments.

---

> > ### Comment · Reviewer_BUmF · 2025-11-27
> >
> > Thank you for your response, and I apologize for the delay. After reading the rebuttal to other reviewers in detail, it raises some additional concerns. When discussing AI-generated text detection, this work overlooks an important line of research (title: Can AI-Generated Text be Reliably Detected?), which examines the reliability of AI-generated text detection. It is essential to demonstrate the reliability of an AI-generated text detector against various types of attacks, such as paraphrasing attacks. Without this, it is challenging to highlight the utility of the detection approach. The authors do mention adversarial attacks in the conclusion, but I was unable to locate similar experiments in the main body. I apologize if I missed anything, but I am updating my scores for now in this light, and will look for further response from the authors to revise my scores again.
> >
> > Thank you.

---

> > > ### Author Response · Authors · 2025-11-27
> > > **A Crucial Clarification**
> > >
> > > Thank you again for the helpful comments.
> > >
> > > We believe the root of this issue is a misunderstanding which we can resolve through discussion. In particular, we would like to clarify that DMAP is not a method for detecting machine generated text, and we do not claim such a method. Thus we don't have anything to run paraphrasing experiments against. We do completely agree that, if an algorithm was built on top of DMAP to detect machine generated text, it would be important to analyse paraphrasing attacks.
> > >
> > > We stress again, the utility of DMAP is in its ability to easily see how a text fits within the next token probability distribution of a language model. We have used this to reveal data errors in NLP work, demonstrate that the fundamental probability curvature hypothesis in detecting machine generated text is often wrong, and suggest a role of temperature-sampling in the over-confidence problem for instruction tuned models.
> > >
> > > Thanks again for your careful consideration of our paper. Please do let us know if there's anything else that we can clarify.

---

> ### Author Response · Authors · 2025-11-28
> **Further experiments added**
>
> Thanks again for your comments. After some further thought on the topic of paraphrasing you raised, we realised there was an interesting experiment to be run related to paraphrasing. We have added an additional appendix with 3 new figures (See Appendix I, Figures 13,14,15), and included the citation you suggested.
>
> While DMAP is not itself a text detector as discussed above, your feedback highlighted it would still be very interesting to ask the question: what do DMAP plots of paraphrased text look like?
>
> These new experiments answer this question, and show that DMAP plots clearly distinguish between human generated and machine-generated paraphrased text. This suggests a very promising line of work would be to build a specialised machine-generated text detector on top of DMAP, as it would likely be far more robust to paraphrasing attacks than existing methods. In addition, the DMAP visualisations shed light on subtle differences between machine-generated and paraphrased machine-generated text - helping us understand as a community what exactly paraphrasing attacks are doing at a statistical level.
>
> Once more we’d like to thank you for your comments, as we would not have thought to run this experiment without your input and found the results to be very interesting and a valuable contribution to the paper.

---

### Author Response · Authors · 2025-11-25
**Response to all referees**

Dear Referees,

many thanks for your careful reading of our manuscript, we are very happy to have received such a considered set of reviews and helpful comments. We have uploaded a revised version of our work.

As promised, we have released our code anonymously here: https://anonymous.4open.science/r/dmap-3B45/README.md [anonymous.4open.science]. At the root of the repository is a python package implementing DMAP with a simple interface, intended to enable the community to easily make use of the tool. An interactive demo is also available in the notebook file demo.ipynb. In the “review/" directory you will find our raw research code used to generate the empirical results found in this paper.

---

### Author Response · Authors · 2025-12-02
**Summary of the Discussion Period**

To aid the final stages of the review process, this comment collects the improvements and additional experiments we added based on reviewer feedback.

&nbsp;

### Summary

- Positive reviews (6, 8): all minor requests fully addressed (Appendices E, F and I).
- Critical reviews (4, 4): raised addressable technical questions directly resolved with substantial new experiments. These were:
  - Robustness to prompt inclusion/exclusion (5pHM): resolved with Appendix G (Figures 8, 9, 10 and 11, 50 experiments). **Reviewer 5pHM explicitly stated they would increase their score** upon seeing this evidence.
  - Comparison with the probability integral transform (PIT) baseline (soHE): resolved with Appendix H (Figure 12, 8 experiments).

Below, we briefly break down our responses by reviewer. In total, we added 5 new appendices containing 11 Figures and 94 individual experiments to comprehensively address all raised concerns. We thank the reviewers for their helpful feedback.

&nbsp;

### Further details

**Reviewer BUmF (Original score: 6)**

- Added Appendix I with 9 plots (Figures 13, 14 and 15) demonstrating the robustness of DMAP to paraphrasing attacks.
- Added Appendix F with 3 plots (Figure 7) and a discussion motivating the entropy-weighted variant of DMAP.
- Additional clarifications in the introduction and throughout the manuscript as requested.

**Reviewer bpqg (Original score: 8)**
- Added Appendix E with a discussion on algorithmic efficiency and 24 additional plots (Figures 5 and 6) that empirically analyze the convergence of DMAP histograms with respect to the number of tokens.
- Significantly improved the illustrative diagram (Figure 1).
- Enhanced mathematical formulations throughout the manuscript.

**Reviewer soHE (Original score: 4)**
- Added Appendix H with 8 plots (Figure 12) that empirically compare DMAP to the probability integral transform (PIT). These plots show that PITs are not a suitable replacement for DMAP. In particular, these experiments show dynamic reordering of tokens in DMAP is critical for enabling the variety of applications we discuss.
- Added a dedicated paragraph in the related work section discussing the relationship of DMAP to PITs.
- Appendix E (mentioned above) contains convergence analysis addressing efficiency concerns.
- Additional clarifications on how existing metrics and DMAP differ in their use of contextual information.

**Reviewer 5pHM (Original score: 4)**

This reviewer explicitly stated they would increase their score if we demonstrated that DMAP was robust to inclusion or exclusion of the prompt:

*"Does the method still at least give interesting empirical results without access to the prompt, but only the model response? Evidence in this direction would cause me to increase my score."*

- In response, we added Appendix G with 50 plots (Figures 8, 9, 10 and 11), which contains a thorough analysis of whether DMAP is sensitive to inclusion of the prompt. In short, this appendix shows DMAP retains equal utility with or without access to the prompt, since exclusion of the prompt at typical sample sizes introduces only negligible noise.
- Clarified and elaborated on DMAP applications beyond text detection (as shown in Section 5 of our manuscript), including:
  - Improving robustness and reproducibility in LLM research.
  - Detecting the impact of post-training with synthetic data on downstream frontier models.

---

### Meta-Review · Area_Chair_TQkk · 2026-01-06

**Summary:**

The paper introduces DMAP (Distribution Map), a visualization and statistical tool that maps text through a language model to samples in the unit interval [0,1] to analyze next-token probability distributions. The authors use this method to uncover significant data errors in prior AI detection literature (specifically regarding the confusion between top-k and pure sampling) and to investigate instruction-tuning artifacts like overconfidence. Reviewers initially raised valid concerns regarding the method's novelty compared to the Probability Integral Transform (PIT), its robustness in practical scenarios (missing prompts, paraphrasing), and efficiency. The rebuttal phase was exceptionally productive, with the authors adding appendices and new experiments to address these specific points.

**Reviewer Concerns:**

I believe the authors have addressed the majority of concerns, turning potential rejection grounds into strengths of the paper.
- Robustness without Prompts (Reviewer 5pHM): This was a critical blocker. The reviewer questioned if the method works when the prompt is unavailable (a common real-world detection scenario). The authors added Appendix G (Figures 8-11), empirically demonstrating that for texts of reasonable length (N>300), DMAP functions effectively without access to the prompt.
- Comparison to Probability Integral Transform (Reviewer soHE): The reviewer argued the method was similar to PIT. The authors added Appendix H (Figure 12) to empirically demonstrate that standard PIT (with random ordering) fails to capture the signal that DMAP’s dynamic reordering captures, thereby justifying their specific methodological choices.
- Robustness to Paraphrasing (Reviewer BUmF): During the discussion, this reviewer raised a new concern about paraphrasing attacks. In response, the authors ran new experiments using the DIPPER model (Appendix I, Figures 13-15), showing DMAP can distinguish paraphrased machine text from human text.
- Efficiency and Convergence (Reviewer bpqg): The reviewer requested analysis on sample complexity. The authors added Appendix E with empirical convergence plots to address this.

**Reviewer Scores:**

- Reviewer 5pHM (Original: 4): Likely to increase (to 6). This reviewer explicitly stated in their review: "Evidence in this direction [robustness without prompts would cause me to increase my score" The authors provided exactly this evidence in Appendix G.
- Reviewer BUmF (Original: 6): This reviewer was already leaning positive but raised the paraphrasing issue late in the process, stating they would update their score "in this light". The authors provided the requested paraphrasing experiments in Appendix I.
- Reviewer soHE (Original: 4): Likely to increase (to 6). The authors directly addressed the "PIT similarity" critique with empirical comparisons showing why DMAP is distinct and superior for this task. While the reviewer may still see it as a variant of PIT, the utility is now better proven.
- Reviewer bpqg (Original: 8): Likely to stay 8. The reviewer was already very positive, and the authors addressed their minor concerns regarding math presentation and efficiency plots.

---

### Decision · Program_Chairs · 2026-01-26

Accept (Poster)